# Bayesian optimization with derivatives acceleration

**Guillaume Perrin**                                                     *guillaume.perrin@univ-eiffel.fr*
*Université Gustave Eiffel, COSYS, 5 Bd Descartes*
*77454 Marne-La-Vallée, France*

**Rodolphe Leriche**                                                              *leriche@emse.fr*
*LIMOS (CNRS, Mines Saint-Etienne, UCA)*
*Saint-Etienne, France*

**Reviewed on OpenReview:** *https: // openreview. net/ forum? id= XXXX*

## Abstract

Bayesian optimization algorithms form an important class of methods to minimize functions that are costly to evaluate, which is a very common situation. These algorithms iteratively infer Gaussian processes from past observations of the function and decide where new observations should be made through the maximization of an acquisition criterion. Often, the objective function is defined on a compact set such as in a hyper-rectangle of the $d$-dimensional real space, and the bounds are chosen wide enough so that the optimum is inside the search domain. In this situation, this work provides a way to integrate in the acquisition criterion the *a priori* information that these functions, once modeled as GP trajectories, should be evaluated at their minima, and not at any point as usual acquisition criteria do. We propose an adaptation of the widely used Expected Improvement acquisition criterion that accounts only for GP trajectories where the first order partial derivatives are zero and the Hessian matrix is positive definite. The new acquisition criterion keeps an analytical, computationally efficient, expression. This new acquisition criterion is found to improve Bayesian optimization on a test bed of functions made of Gaussian process trajectories in low dimension problems. The addition of first and second order derivative information is particularly useful for multimodal functions.

## 1 Introduction

Over the last 20 years, Bayesian optimization (BO) methods have established themselves as one of the references for approximating the point(s) minimizing an expensive-to-evaluate black-box function, from as few calls to this function as possible. This is reflected in the existence of many reviews and tutorials on BO in the literature (see for instance Jones (2001); Sobester et al. (2008); Shahriari et al. (2015); Gramacy (2020); Garnett (2023); Frazier (2018), as well as many applications of BO in industrial applications, such as aeronautics Forrester et al. (2007); Lam et al. (2018) or agriculture Picheny et al. (2017)). Today, the machine learning community is a key contributor to BO advances, motivated by the need to optimize hyper-parameters Bergstra et al. (2011); Snoek et al. (2012); Klein et al. (2017); Wu et al. (2019); Turner et al. (2021) or exploration strategies in reinforcement learning Wang et al. (2023). More specifically, BO is concerned with minimization problems that can be written in the following form:

$$\boldsymbol{x}^{\star} \in \underset{\boldsymbol{x} \in \mathbb{X}}{\arg \min }\, y(\boldsymbol{x}), \tag{1}$$

where $y$ is a pointwise observable function defined over the compact set $\mathbb{X} \subset \mathbb{R}^d$, $d \geqslant 1$. BO assumes that $y$ can be usefully represented as a Gaussian process (GP), whose mean and covariance functions have been identified from a limited number of calls to function $y$. It then sequentially adds new observations of $y$ at points maximizing an *acquisition criterion* whose objective, in the search for the global minimum, is

to make a judicious trade-off between the exploration of $\mathbb{X}$ and the exploitation of past observations. In the theory of decision under uncertainty, acquisition criteria are the expectation of a *utility* of the possible function observations according to the stochastic model of the objective function Žilinskas & Calvin (2019).

Several acquisition criteria have been proposed. The earliest, one-dimensional, version of BO Kushner (1962) involved the probability of improvement and an upper confidence bound. The upper confidence bound was later theoretically studied in many dimensions in Srinivas et al. (2010). Another early BO acquisition criterion was described in Močkus (1972) which is, since Frazier & Powell (2007), called the knowledge gradient. It is a one-step-ahead expected progress in GP mean. The Expected Improvement beyond the current best observation (EI) is the most classical acquisition criterion. The EI has a simple interpretation and an analytical expression deprived of parameters to tune, two features which have contributed to its popularity. It was first proposed in Saltenis (1971) and popularized in Schonlau (1997); Jones et al. (1998); Močkus (2012). More recently, acquisition criteria based on information theory have been suggested which target entropy reductions in the GP model extrema Hernández-Lobato et al. (2014) or locations of extrema Villemonteix et al. (2009); Hennig & Schuler (2012).

BO is particularly efficient when the dimension of the search space remains limited ($d \leqslant 5$ to $10$) and when the function is multimodal with some structure Le Riche & Picheny (2021). Several adaptations of this formalism have been proposed to extend the efficiency of these approaches to larger input spaces, by playing directly on the acquisition criterion Siivola et al. (2018), on the identification of latent spaces of reduced dimensions Bouhlel et al. (2016); Gaudrie et al. (2020), on the introduction of trust regions Diouane et al. (2022), or by replacing the GP by a Bayesian neural network Kim et al. (2022).

It sometimes happens that the derivatives of the true function at the observed points is available (e.g., through automatic differentiation, or adjoint codes in partial differential equations solvers). It is then possible to add these derivatives as part of the observations of a vectorized Gaussian process composed of the function prediction and its derivatives Laurent et al. (2019). All the above acquisition criteria could then be calculated with such a gradient-enriched underlying Gaussian Process. This has been done with the gradient-knowledge criterion in Wu et al. (2017).

It is nevertheless interesting to note that all of these methods only exploit a limited part of the information conveyed by the GP (once conditioned by the observations of the true function and potentially by the observations of its gradient). In particular, they do not take into account the information that the GP derivatives could bring, whether the function $y$ is convex or not, and even if the derivatives of $y$ are not observed. Indeed, when $y$ is twice differentiable, it is well known that the first derivatives of $y$ become zero and that its Hessian matrix is positive definite at its minimum (unless the minimum lies at an edge of the domain). It is reasonable to believe that the minimization strategy can only benefit from this supplementary knowledge on derivatives. Figure 1 provides a graphical illustration to support this intuition. It shows three plots with GP trajectories conditioned by three observations (also known as a kriging model). The two bottom plots further condition the trajectories on their derivatives so that they have local minima at the wrong (left) or right location. Because it is easier to force local minima where the true function really has an optimum, the information on local minima helps better discriminating the optimal from the non-optimal region. Note also that in Figure 1, while the first and second order derivatives of the GP trajectories are constrained, no information about the derivatives of the true function is used. This is a key difference with other work on BO with gradient knowledge such as Wu et al. (2017).

With this in mind, the main contribution of this paper is to propose an adaptation of the famous EI criterion so that it integrates the information of zero derivative and positive definite Hessian matrix of the GP trajectories. In other terms, this new criterion only accounts for possible minima of the GP trajectories, as opposed to the traditional EI that can confer a utility to any part of a trajectory. We emphasize that the proposed criterion does not imply that derivatives of the true function $y(\boldsymbol{x})$ be calculated. The derivatives only concern the GP.

The new criterion is meaningful if the minimum is located inside the search domain, which is a reasonable assumption in most applications where, precisely, the bounds are chosen as extremes that should not be reached. A complementary idea for cases where the bounds might be active is nevertheless given as a

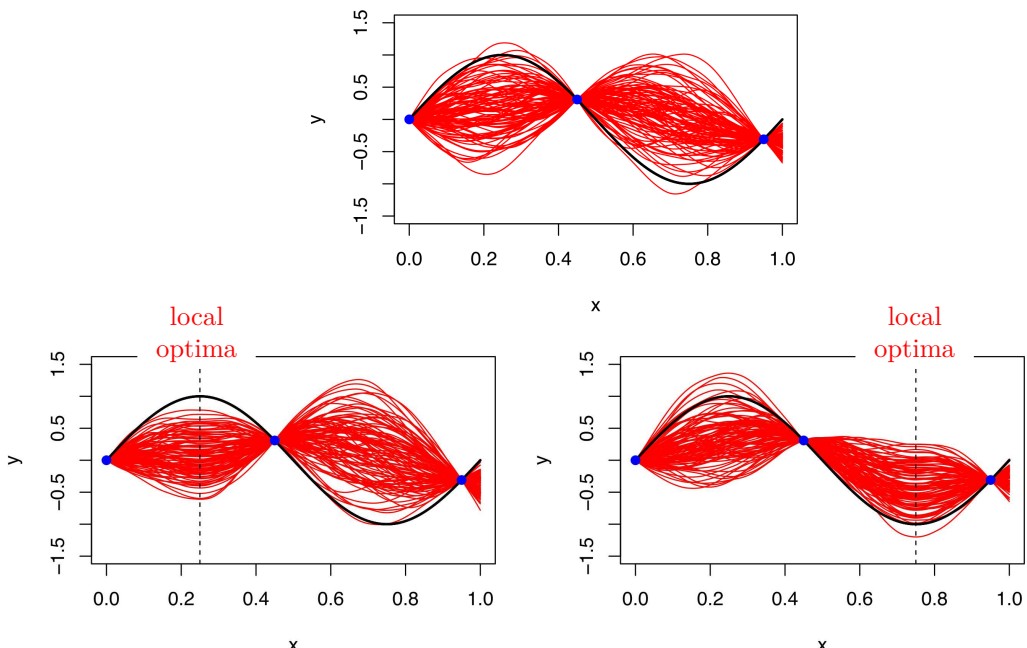

Figure 1: An illustration of the motivation for conditioning kriging trajectories with derivatives. Top: kriging trajectories in red, true function as a black solid line. Bottom: kriging trajectories forcing, by Gaussian conditioning, a zero derivative and a positive second derivative at the vertical dotted bar, i.e. at the global maximum of the true function in the figure on the left, and at the global minimum in the figure on the right. The difference between trajectories at the maximum and minimum of the true function is more apparent when forcing local minima at the right location.

perspective: a method is proposed to estimate the likelihood that the minimum of $y$ is on the edge of the domain.

Empirically investigating the effect of a new idea – here adding derivatives acceleration – on an optimization algorithm is difficult because the performance of an algorithm depends on both the function it is applied to and the tuning of its hyperparameters. The empirical tests we provide are designed to exclusively show the effects of the derivatives acceleration while avoiding all such experimental side effects. This is achieved firstly by testing on Gaussian processes whose hyper-parameters are known, therefore guaranteeing the compatibility of the model and the test function. Secondly, the maximization of the acquisition criteria is done with extreme care, which is feasible up to dimension 5 and needs to be relaxed beyond.

The outline of this paper is as follows. Section 2 recalls the theoretical bases of the Gaussian process regression (GPR) and its use for the minimization of black-box functions. Section 3 introduces the acquisition criterion we propose for taking into account information on the derivatives of $y$. Section 4 then illustrates the benefits of this new acquisition criterion on simulated test functions that can be modeled as realizations of Gaussian processes. Section 5 describes how optima on the bounds can be handled and concludes the paper.

## 2   The BO general framework

For $d \geqslant 1$, let $\mathbb{X}$ be a compact subset of $\mathbb{R}^d$. In this work, we are interested in finding the solution(s) $\boldsymbol{x}^\star$ of the optimization problem defined by Eq. (1) using as few pointwise observations of $y$ as possible. Anticipating the developments in the following sections exploiting the gradient of $y$, we assume that $y$ is an element of $\mathcal{C}^2(\mathbb{X}, \mathbb{R})$, the set of real-valued twice continuously differentiable functions defined on $\mathbb{X}$. In addition, we treat $\mathbb{X}$ as explicit, which means that the function $y$ cannot be evaluated outside the search region (it is defined as a product of intervals in the applications).

To solve this problem, we consider Bayesian Optimization guided by the Expected Improvement (EI) acquisition criterion. Such methods are sometimes called Efficient Global Optimization algorithms in reference to Jones et al. (1998), although implementations (of the GP and of the EI maximization) vary. The choice of the EI acquisition criterion is guided by simplicity: it is the most standard criterion and most importantly, it does not require GP simulations to be evaluated. Others criteria could benefit from derivatives acceleration as discussed in the perspectives of this article (Section 5.2).

BO relies on the evaluation of the objective function at a sequence of well-chosen points as summarized hereunder and in Algorithm 1.

**Initialization**

To begin, the function $y$ is evaluated at $N_0$ points uniformly chosen in $\mathbb{X}$ (typically according to a space-filling design of experiments (DoE) Fang et al. (2006); Perrin & Cannamela (2017)). We note $(\boldsymbol{x}^{(n)}, y_n := y(\boldsymbol{x}^{(n)}))_{n=1}^{N_0}$ the obtained pairs. Given this available data, a GP-based surrogate model is trained for $y$. To obtain convergence results, a common theoretical assumption is that $y$ is a particular realization of a Gaussian process $Y \sim \mathrm{GP}(\mu, C)$, whose prior mean and prior covariance functions are noted $\mu$ and $C$ respectively (see Santner et al. (2003); Rasmussen (2003) for more details about Gaussian process regression). In practice, it is only required that $y$ can be observed at a finite number of points and the assumption of $y$ being a sample of $Y$ may not hold. The algorithm then conditions $Y$ to interpolate the $N_0$ available input-output pairs, resulting in a new $Y_{N_0} \sim \mathrm{GP}(\mu_{N_0}, C_{N_0})$, where:

$$\mu_{N_0}(\boldsymbol{x}) = \mu(\boldsymbol{x}) + C(\boldsymbol{x}, \boldsymbol{X})C(\boldsymbol{X}, \boldsymbol{X})^{-1}(y(\boldsymbol{X}) - \mu(\boldsymbol{X})), \quad \boldsymbol{x} \in \mathbb{X}, \tag{2}$$

$$C_{N_0}(\boldsymbol{x}, \boldsymbol{x}') = C(\boldsymbol{x}, \boldsymbol{x}') - C(\boldsymbol{x}, \boldsymbol{X})C(\boldsymbol{X}, \boldsymbol{X})^{-1}C(\boldsymbol{X}, \boldsymbol{x}'), \quad \boldsymbol{x}, \boldsymbol{x}' \in \mathbb{X}. \tag{3}$$

In the former expressions, $\boldsymbol{X} := [\boldsymbol{x}^{(1)} \cdots \boldsymbol{x}^{(N_0)}]^T$ is the $(N_0 \times d)$-dimensional matrix that gathers the available input points, and for any function $f$ and $g$ defined on $\mathbb{X}$ and $\mathbb{X} \times \mathbb{X}$ respectively, the following notation is adopted:

$$(f(\boldsymbol{X}))_n = f(\boldsymbol{x}^{(n)}), \quad (g(\boldsymbol{X}, \boldsymbol{X}))_{nm} = g(\boldsymbol{x}_n, \boldsymbol{x}_m), \quad 1 \leqslant n, m \leqslant N_0. \tag{4}$$

**Iteration**

Given $N \geqslant N_0$ evaluations of $y$, an acquisition criterion is introduced to choose at which point to carry out the $(N + 1)^{\text{th}}$ evaluation of $y$. In the noise-free setting, the classical acquisition criterion is the Expected Improvement (EI). It is the expectation of a utility at $\boldsymbol{x}$ defined as the progress below the current best observation:

$$\begin{aligned}
\text{EI}_N(\boldsymbol{x}) &:= \mathbb{E}\left[\max(0, y_{\min} - Y_N(\boldsymbol{x}))\right] = \int_{\mathbb{R}} \max(0, y_{\min} - y) f_{Y_N(\boldsymbol{x})}(y) dy \\
&= \sigma_N(\boldsymbol{x})\left(U_N(\boldsymbol{x})\Phi\left(U_N(\boldsymbol{x})\right) + \phi\left(U_N(\boldsymbol{x})\right)\right).
\end{aligned} \tag{5}$$

Here, $U_N(\boldsymbol{x}) := (y_{\min} - \mu_N(\boldsymbol{x}))/\sigma_N(\boldsymbol{x})$, $\sigma_N(\boldsymbol{x}) := \sqrt{C_N(\boldsymbol{x}, \boldsymbol{x})}$, $y_{\min}$ is the current minimum of the $N$ observations of $y$, noted $y(\boldsymbol{x}^{(1)}), \ldots, y(\boldsymbol{x}^{(N)})$, $\Phi$ and $\phi$ denote the probability density function (PDF) and cumulative density function (CDF) of the standard Gaussian variables, and $f_{Y_N(\boldsymbol{x})}(y) = \phi((y - \mu_N(\boldsymbol{x}))/\sigma_N(\boldsymbol{x}))/\sigma_N(\boldsymbol{x})$ is the PDF of the Gaussian random variable $Y_N(\boldsymbol{x}) \sim \mathcal{N}(\mu_N(\boldsymbol{x}), \sigma_N(\boldsymbol{x})^2)$, where

$$Y_N := Y \mid Y(\boldsymbol{x}^{(1)}) = y(\boldsymbol{x}^{(1)}), \ldots, Y(\boldsymbol{x}^{(N)}) = y(\boldsymbol{x}^{(N)}). \tag{6}$$

By construction, this acquisition criterion seeks a compromise between exploitation (first term) and exploration (second term) for the global search of the minimum, and the next evaluation point is chosen such that

$$\boldsymbol{x}^{(N+1)} \in \underset{\boldsymbol{x} \in \mathbb{X}}{\arg\max}\, \text{EI}_N(\boldsymbol{x}). \tag{7}$$

**Stopping criterion**

For most existing implementations of BO, the stopping criterion is a maximum number of evaluations of $y$. Indeed, unlike gradient-based approaches for minimizing convex functions, once a local minimum of $y$ has been found, there is no theoretical guarantee that it corresponds to the global minimum of $y$. While it may be tempting, stopping the search when the expected improvement drops below a lower bound is unstable in practice as the EI changes a lot with the GP length scales.

**Degrees of freedom**

The performance of the BO method depends on several degrees of freedom that vary between implementations. The choice for $\mu$ and $C$, the way the parameters on which $\mu$ and $C$ depend are optimized, the ratio $N_0/\text{budget}$, the way the initial DoE is constructed, the way the acquisition criterion is maximized are all important (see Le Riche & Picheny (2021) for an investigation of the influence of these choices).

However, as the paper studies an adaptation of the acquisition criterion, it is clearer to fix these parameters to standard values of the literature. To this end, the function $\mu$ is taken as a constant, and the function $C$ is chosen in the class of tensorized Matérn kernels with smoothing parameter $\nu = 5/2$ (see Santner et al. (2003) for alternative classes of functions):

$$\mu(\boldsymbol{x}) := \beta, \quad C(\boldsymbol{x}, \boldsymbol{x}') := \sigma^2 \prod_{i=1}^{d} \kappa\left(\frac{|x_i - x_i'|}{\ell_i}\right), \quad \boldsymbol{x}, \boldsymbol{x}' \in \mathbb{X}, \tag{8}$$

---

**Algorithm 1:** Standard BO algorithm.

---

Choose $N_0$, budget, $Y \sim \mathrm{GP}(\mu, C)$ ;
$\rightarrow$ **Initialization**
Draw at random $N_0$ points $\boldsymbol{x}^{(1)}, \ldots, \boldsymbol{x}^{(N_0)}$ in $\mathbb{X}$ ;
Compute $y(\boldsymbol{x}^{(n)})$, $1 \leqslant n \leqslant N_0$, estimate the parameters on which $\mu$ and $C$ depend ;
Define $Y_{N_0} := Y | Y(\boldsymbol{x}^{(1)}) = y(\boldsymbol{x}^{(1)}), \ldots, Y(\boldsymbol{x}^{(N_0)}) = y(\boldsymbol{x}^{(N_0)})$ ;
Set $k = 0$ ;
$\rightarrow$ **Iteration**
**while** $k <$ *budget* **do**
    Search for $\boldsymbol{x}^{(N_0+k+1)} := \arg\max_{\boldsymbol{x} \in \mathbb{X}} \mathrm{EI}_{N_0+k}(\boldsymbol{x})$ ;
    Evaluate $y$ at $\boldsymbol{x}^{(N_0+k+1)}$ (and potentially adjust the expressions of $\mu$ and $C$) ;
    Define $Y_{N_0+k+1} := Y | Y(\boldsymbol{x}^{(1)}) = y(\boldsymbol{x}^{(1)}), \ldots, Y(\boldsymbol{x}^{(N_0+k+1)}) = y(\boldsymbol{x}^{(N_0+k+1)})$ ;
    Update $k \leftarrow k + 1$
**end**
Return $\min_{1 \leqslant i \leqslant N_0+\text{budget}} y(\boldsymbol{x}^{(i)})$.

---

$$\kappa(u) := \left(1 + \sqrt{5}u + \frac{5}{3}u^2\right) \exp\left(-\sqrt{5}u\right), \quad u \geqslant 0. \tag{9}$$

The Matérn 5/2 kernel is selected here because it is well-performing and common in the literature (Le Riche & Picheny (2021)). Furthermore, we will soon introduce an acquisition criterion that needs the fourth order derivatives of the kernel (to have information about the curvatures of the trajectories). The Matérn 5/2 kernel is precisely four times differentiable, yielding trajectories that are two times continuously differentiable. The hyperparameter vector $\boldsymbol{\theta} := (\beta, \sigma, \ell_1, \ldots, \ell_d)$ will either be considered known (via the definition of test functions to be minimized in the form of a particular realization of a Gaussian process of chosen parameters), or estimated by its maximum likelihood estimator (see Williams & Rasmussen (2006) for further details). As we focus on costly functions, we will set the maximal budget between 10 and 20 times the dimension $d$ of the problem, while $N_0$ will be chosen small (most of the time we will have $N_0 = 3$). The initial DoE will always be a random space-filling Latin Hypercube Sample (LHS) Damblin et al. (2013); Perrin & Cannamela (2017). For objective numerical comparisons, the maximization of the acquisition criteria, whether it is the EI in Equation (7) or one of the new criteria of Section 3, is always carried out in the same way. At each iteration, the acquisition criterion is first evaluated at a very large number of points randomly chosen in $\mathbb{X}$ (typically of the order of $10^{d+1}$). The Nelder-Mead algorithm Nelder & Mead (1965) then maximizes the acquisition criterion starting from the 10 most promising points among the random points.

## 3 Extending the Expected Improvement with derivatives

We now show how to extend the Expected Improvement acquisition criterion so that it accounts for gradient and Hessian information. The principles underlying the calculations are that GP derivatives are GPs, and that local optima away from the bounds coincide with canceling derivatives and positive definite Hessians. These principles have already been used in the context of BO in Hernández-Lobato et al. (2014) for approximating the entropy of local optima. An independent and differing version, adapted to EI, is described hereafter.

### 3.1 Reminders on Gaussian process derivation

The acquisition criteria reviewed in the Introduction, in particular the EI, are only based on the distribution of $Y_N(\boldsymbol{x})$ and do not include information related to higher derivatives. Yet, when the functions $\boldsymbol{x} \mapsto \mu(\boldsymbol{x})$ and $(\boldsymbol{x}, \boldsymbol{x}') \mapsto C(\boldsymbol{x}, \boldsymbol{x}')$ are sufficiently regular, the statistical properties of the derivatives of $Y$ can be deduced by simple derivations of $\mu$ and $C$. Indeed, as the Gaussian distribution is stable by linear

operations, for any linear operator $\mathcal{L}$ such that $\mathcal{L}y$ is a function from $\mathbb{R}^d$ to $\mathbb{R}^{d_\mathcal{L}}$, $\mathcal{L}Y$ is also a Gaussian process, with:

$$\mathbb{E}\left[\mathcal{L}Y(\boldsymbol{x})\right] = \mathcal{L}\mu(\boldsymbol{x}), \quad \text{Cov}(\mathcal{L}Y(\boldsymbol{x}), \mathcal{L}Y(\boldsymbol{x}')) = \mathcal{L}C(\boldsymbol{x}, \boldsymbol{x}')\mathcal{L}^T. \tag{10}$$

Here, the notations $\mathcal{L}C(\boldsymbol{x}, \boldsymbol{x}')$ and $C(\boldsymbol{x}, \boldsymbol{x}')\mathcal{L}^T$ indicate that operator $\mathcal{L}$ is applied as a function of $\boldsymbol{x}$ and $\boldsymbol{x}'$ respectively, so that $\text{Cov}(\mathcal{L}Y(\boldsymbol{x}), \mathcal{L}Y(\boldsymbol{x}'))$ is a $(d_\mathcal{L} \times d_\mathcal{L})$-dimensional matrix. In particular, for $d_\mathcal{L} = 1 + d(d+3)/2$, if we choose

$$\mathcal{L} : Y \mapsto \mathcal{L}Y := \left( Y, \frac{\partial Y}{\partial x_1}, \dots, \frac{\partial Y}{\partial x_d}, \frac{\partial^2 Y}{\partial x_1^2}, \dots, \frac{\partial^2 Y}{\partial x_1 \partial x_2}, \dots, \frac{\partial^2 Y}{\partial x_d^2} \right),$$

we obtain the joint distribution of $Y$ and its first and second order derivatives. For each twice-differentiable function $z$, we introduce the following notations,

$$\partial z := \begin{bmatrix} \frac{\partial z}{\partial x_1} \\ \vdots \\ \frac{\partial z}{\partial x_d} \end{bmatrix}, \quad \partial^2 z := \begin{bmatrix} \frac{\partial^2 z}{\partial x_1^2} & \cdots & \frac{\partial^2 z}{\partial x_1 \partial x_d} \\ \vdots & \ddots & \vdots \\ \frac{\partial^2 z}{\partial x_1 \partial x_d} & \cdots & \frac{\partial^2 z}{\partial x_d^2} \end{bmatrix}, \quad D^2 z := \text{diag}(\partial^2 z) = \begin{bmatrix} \frac{\partial^2 z}{\partial x_1^2} \\ \vdots \\ \frac{\partial^2 z}{\partial x_d^2} \end{bmatrix}, \tag{11}$$

and we denote by $\mathcal{M}^+(d)$ the set of $(d \times d)$-dimensional positive definite matrices.

### 3.2 An acquisition criterion accounting for the derivatives

For any $\boldsymbol{x}$ in $\mathbb{X}$, it is well known that if $\partial z(\boldsymbol{x}) = \boldsymbol{0}$ and $\partial^2 z(\boldsymbol{x}) \in \mathcal{M}^+(d)$, $\boldsymbol{x}$ is a local minimum of $z$. As the input space $\mathbb{X}$ is bounded, the reciprocal is however not true, since a local minimum can be found at the boundary of $\mathbb{X}$ with a non-zero gradient and/or $\partial^2 z(\boldsymbol{x}) \notin \mathcal{M}^+(d)$. The case when the optima are on the bounds will be discussed in Section 5.1. For now we focus on the interior of $\mathbb{X}$, to integrate as prior knowledge that the gradient will be zero and the matrix of curvatures positive definite at the local minima of $y$, the EI criterion defined by Eq. (5) can be replaced by:

$$\text{deriv-EI}_N(\boldsymbol{x}) := \mathbb{E}\left[ \mathbb{1}_{\boldsymbol{R}_N(\boldsymbol{x})\partial Y_N(\boldsymbol{x}) \in \mathcal{B}(\varepsilon), \partial^2 Y_N(\boldsymbol{x}) \in \mathcal{M}^+(d)} \max(0, y_{\min} - Y_N(\boldsymbol{x})) \right], \tag{12}$$

where for any $\varepsilon > 0$, $\mathcal{B}(\varepsilon) := \left\{ \dot{\boldsymbol{y}} \in \mathbb{R}^d, \ \|\dot{\boldsymbol{y}}\| \leqslant \varepsilon \right\}$ is the $d$-dimensional hypersphere of radius $\varepsilon$, $\boldsymbol{R}_N(\boldsymbol{x})$ is a matrix such that

$$\boldsymbol{R}_N(\boldsymbol{x})\text{Cov}(\partial Y_N(\boldsymbol{x}))\boldsymbol{R}_N(\boldsymbol{x})^T = \boldsymbol{I}_d, \tag{13}$$

and for any event $a$, $\mathbb{1}_a$ is equal to 1 if $a$ is true and to 0 otherwise. $\partial Y_N(\boldsymbol{x})$ has a covariance matrix (made of the second derivatives of $C_N(\boldsymbol{x}, \boldsymbol{x}')$) with correlations and its density has ellipsoidal level sets. The normalized vector $\boldsymbol{R}_N(\boldsymbol{x})\partial Y_N(\boldsymbol{x})$ is uncorrelated, its covariance is the identity $\boldsymbol{I}_d$, and its density has spherical level sets that can be compared to the sphere $\mathcal{B}(\epsilon)$. The dependency of this scaling on $\boldsymbol{x}$ disappears for $\varepsilon$ small (see Appendix A). Equivalently, we can write the criterion deriv-EI$_N$ as

$$\text{deriv-EI}_N(\boldsymbol{x}) := \mathbb{E}\left[ \mathbb{1}_{\partial Y_N(\boldsymbol{x}) \in \mathcal{E}(\boldsymbol{x}, \varepsilon), \partial^2 Y_N(\boldsymbol{x}) \in \mathcal{M}^+(d)} \max(0, y_{\min} - Y_N(\boldsymbol{x})) \right], \tag{14}$$

with $\mathcal{E}(\boldsymbol{x}, \varepsilon) := \left\{ \dot{\boldsymbol{y}} \in \mathbb{R}^d, \ \dot{\boldsymbol{y}}^T \boldsymbol{R}_N(\boldsymbol{x})^T \boldsymbol{R}_N(\boldsymbol{x})\dot{\boldsymbol{y}} \leqslant \varepsilon^2 \right\}$ a $d$-dimensional ellipsoid. In connection with theoretical decision under uncertainty Žilinskas & Calvin (2019), deriv-EI$_N(\boldsymbol{x})$ is the expectation of a utility of the function model (the GP trajectories) at $\boldsymbol{x}$. Here, the utility is defined as the progress of the stochastic model below the best observation knowing that the function model has a minimum at $\boldsymbol{x}$ i.e., it has null first order derivatives and positive curvatures. The key idea of deriv-EI is to account only for minima of

the possible functions. On the contrary, EI accounts for any value of the possible functions which is below the best observation, which is less consistent with the goal of minimization. Because it characterizes the behavior of the minima of the GP realizations, deriv-EI can be seen as a criterion between EI and information theoretic criteria based on the expected reduction in entropy of the optima Hernández-Lobato et al. (2014).

By considering deriv-EI$_N$ rather than EI$_N$ as a new acquisition criterion in Algorithm 1, we expect to improve its exploitation capabilities, without degrading its exploration capabilities too much. Like EI$_N$, deriv-EI$_N$ needs only evaluations of the true function, $y(\boldsymbol{x}^{(1)}), \ldots, y(\boldsymbol{x}^{(N)})$ through $Y_N$, $\partial Y_N$ and $\partial^2 Y_N$. It does not need derivatives of the true function, $y$. Only the GP is differentiated. However, this acquisition criterion can no longer be calculated simply, and in the general case it will require the use of sampling techniques for its evaluation, which may complicate its use. Nevertheless, if we choose $\varepsilon$ small, if we neglect the off-diagonal terms of the Hessian (as it was already proposed in Hernández-Lobato et al. (2014)) while assuming a well-chosen conditional independence of its diagonal terms, we obtain the following relaxed acquisition criterion (see Appendix A for a detailed derivation):

$$\text{deriv-EI}_N(\boldsymbol{x}) \approx \text{LikelyMin}_N(\boldsymbol{x}) \times \text{cond-EI}_N(\boldsymbol{x}), \tag{15}$$

$$\text{LikelyMin}_N(\boldsymbol{x}) := v \times \varepsilon^d \times \exp\left(-\frac{\dot{\boldsymbol{m}}^T \dot{\boldsymbol{S}}^{-1} \dot{\boldsymbol{m}}}{2}\right) \times \prod_{i=1}^{d} \Phi\left(\frac{\ddot{\tau}_i}{\sqrt{1-r_i^2}}\right), \tag{16}$$

$$\text{cond-EI}_N(\boldsymbol{x}) := s\left((z_{\min} - a)\Phi(z_{\min}) + \phi(z_{\min})\right), \tag{17}$$

where $v$ is a constant that does not depend on $\boldsymbol{x}$ and $\varepsilon^d$, and where the following notations have been introduced to simplify the expressions:

$$\partial Y_N(\boldsymbol{x}) \sim \mathcal{N}\left(\dot{\boldsymbol{m}}, \dot{\boldsymbol{S}}\right), \quad D^2 Y_N := \left((\partial^2 Y_N)_{1,1}, \ldots, (\partial^2 Y_N)_{d,d}\right), \tag{18}$$

$$(Y_N(\boldsymbol{x}), D^2 Y_N(\boldsymbol{x})))|\partial Y_N(\boldsymbol{x}) = \boldsymbol{0} \sim \mathcal{N}\left(\begin{pmatrix} m \\ \ddot{m}_1 \\ \vdots \\ \ddot{m}_d \end{pmatrix}, \begin{bmatrix} s^2 & \rho_{1,1} & \cdots & \rho_{1,d} \\ \rho_{1,1} & \ddot{s}_1 & \ddots & \vdots \\ \vdots & \ddots & \ddots & \rho_{d-1,d} \\ \rho_{d,1} & \cdots & \rho_{d,d-1} & \ddot{s}_d \end{bmatrix}\right), \tag{19}$$

$$z_{\min} := \frac{y_{\min} - m}{s}, \quad r_i := \frac{\rho_{1i}}{s\ddot{s}_i}, \quad \ddot{\tau}_i = \frac{\ddot{m}_i}{\ddot{s}_i}, \quad a = \sum_{i=1}^{d} \frac{r_i}{\sqrt{1-r_i^2}} \frac{\phi\left(\frac{\ddot{\tau}_i}{\sqrt{1-r_i^2}}\right)}{\Phi\left(\frac{\ddot{\tau}_i}{\sqrt{1-r_i^2}}\right)}. \tag{20}$$

The precise choice of $\varepsilon$ has thus no impact.

### 3.3 Comments on the proposed acquisition criterion

**Analysis of the terms in deriv-EI**
Comparing the criteria EI$_N(\boldsymbol{x})$ and deriv-EI$_N(\boldsymbol{x})$, we first notice the presence of the function $\boldsymbol{x} \mapsto \text{LikelyMin}_N(\boldsymbol{x})$, whose role is to concentrate the search of the new point to be evaluated around the points $\boldsymbol{x}$ that are likely to lead to a zero gradient of $y$ (small values of $\dot{\boldsymbol{m}}^T \dot{\boldsymbol{S}}^{-1} \dot{\boldsymbol{m}}$), while favouring the areas of positive second derivatives (high values of $\Phi(\ddot{\tau}_i/\sqrt{1-r_i^2})$ for all $i$). The second function $\boldsymbol{x} \mapsto \text{cond-EI}_N(\boldsymbol{x})$ estimates the expected improvement assuming that the function has a minimum at $\boldsymbol{x}$, and looks particularly like the expression given by Eq. (5). The more the second derivatives of $Y_N$ will be positive in

probability, which translates into large values of $\ddot{\tau}_i$, the more this similarity will be important because, in this case, the constant $a$ tends towards 0. In addition, as the statistical properties of $Y_N(\boldsymbol{x})$, $\partial Y_N(\boldsymbol{x})$ and $D^2 Y_N(\boldsymbol{x})$ are known explicitly, it is important to notice that the evaluation cost of deriv-$\mathrm{EI}_N(\boldsymbol{x})$ is of the same order of magnitude as that of the classical $\mathrm{EI}_N(\boldsymbol{x})$. Importantly, there is no need to use sampling methods to estimate it.

In addition, if $\mu$ is chosen to be constant and $C$ is a stationary covariance kernel (which remains the most common configuration in BO), then $\partial Y(\boldsymbol{x})$ is statistically independent of $Y(\boldsymbol{x})$ and $D^2 Y(\boldsymbol{x})$ for any $\boldsymbol{x}$ in $\mathbb{X}$. In particular, if we focus on the first iteration of the BO procedure ($N = 0$), and put aside the constraint on the Hessian, it can be noted that for any $\varepsilon > 0$ and any $\boldsymbol{x} \in \mathbb{X}$,

$$\mathbb{E}\left[\mathbf{1}_{\boldsymbol{R}_0(\boldsymbol{x})\partial Y(\boldsymbol{x})\in\mathcal{B}(\varepsilon)}\max(0, y_{\min} - Y(\boldsymbol{x}))\right] = p_\varepsilon \times \mathrm{EI}_0(\boldsymbol{x}), \tag{21}$$

where $p_\varepsilon := \mathbb{P}(\boldsymbol{R}_0\partial Y(\boldsymbol{x}) \in \mathcal{B}(\varepsilon))$ is a constant independent of $\boldsymbol{x}$ as the statistical properties of $\partial Y(\boldsymbol{x})$ do not depend on $\boldsymbol{x}$ (stationarity). In that case, $\mathrm{EI}_0$ is very close to deriv-$\mathrm{EI}_0$ (up to the influence of the second derivatives), and maximizing either of these criteria should give close results. Then, the more the process $Y$ is conditioned by observations of $y$, the more $Y$, $\partial Y$ and $\partial^2 Y$ are correlated, and the more chances there are for deriv-$\mathrm{EI}_N(\boldsymbol{x})$ and $\mathrm{EI}_N(\boldsymbol{x})$ to propose different points. After many observations, it is anticipated that the interesting areas from the EI point of view will have low gradients, so that the two criteria should again propose close new evaluation points. The *a priori* interest of the deriv-$\mathrm{EI}_N(\boldsymbol{x})$ criterion thus lies in intermediate values of $N$, in the exploration of the various local minima of $y$, and the search for the smallest zone of $\mathbb{X}$ likely to contain the global minimum of $y$.

At last, when the dimension $d$ increases, one of the classical difficulties of BO based on $\mathrm{EI}_N$ is to favor exploration over exploitation, by placing a very large number of points on the edges of the domain, which effectively represent the majority of the volume of $\mathbb{X}$ when $d$ is large Siivola et al. (2018). This effect should be limited by substituting deriv-$\mathrm{EI}_N$ for $\mathrm{EI}_N$, i.e., by requiring that each partial derivative of $Y_N$ be close to 0 and that each main curvature be positive through the factor LikelyMin$(\boldsymbol{x})$, which becomes more restrictive as $d$ increases.

**A more exploratory deriv-EI**
In return, by trying to quickly visit potential high-performance local minima, it is possible that the deriv-$\mathrm{EI}_N$ criterion explores fewer regions of $\mathbb{X}$ than $\mathrm{EI}_N$, which could be penalizing for the minimization of functions with multiple local minima. If this were the case (this kind of phenomenon was not observed on the test cases studied in Section 4), several techniques could be proposed to rebalance the exploration/exploitation ratio. For instance, the control of the exploration-exploitation balance by changing target values has been studied in Jones (2001) for the probability of improvement and a likelihood criterion. Such a shift in target around $y_{\min}$ was included in the EI criterion in Berk et al. (2019); Lizotte (2008). Another way of reinforcing exploration with respect to exploitation consists in maximizing the expected improvement at a certain power $p \geqslant 1$. Indeed, by taking $p$ greater than 1, we further encourage low-probability high improvements compared to more probable small improvements. This idea was pursued in Schonlau et al. (1998) where expressions for the generalized $\mathrm{EI}_N$ criterion with $p \geqslant 2$ can be found. The new EI with derivatives can also benefit from elevating the improvement at a given power. It becomes,

$$\text{deriv-}\mathrm{EI}_N^{(p)} := \mathbb{E}\left[\mathbf{1}_{\partial Y(\boldsymbol{x})\in\mathcal{E}(\boldsymbol{x},\varepsilon),\partial^2 Y_N(\boldsymbol{x})\in\mathcal{M}^+(d)}\max(0, y_{\min} - Y_N(\boldsymbol{x}))^p\right].$$

For $p = 2$ (see Appendix A for more details) and using the same notations as in Section 3.2, such a criterion can again be approximated under an analytical form close to the one of Eq. (15):

$$\text{deriv-}\mathrm{EI}_N^{(2)}(\boldsymbol{x}) \approx \text{LikelyMin}(\boldsymbol{x}) \times \text{cond-}\mathrm{EI}^{(2)}(\boldsymbol{x}), \tag{22}$$

$$\text{cond-}\mathrm{EI}^{(2)}(\boldsymbol{x}) := s^2\left((1 + z_{\min}^2 - 2az_{\min})\Phi(z_{\min}) + (z_{\min} - 2a)\phi(z_{\min})\right). \tag{23}$$

**Adaptation to noisy outputs**

The criteria $\text{EI}_N$ and deriv-$\text{EI}_N$ are introduced in a context where the observations of $Y$ are assumed noise-free. If it turns out that these outputs are in fact noisy, and if this noise can be modeled as a centered Gaussian vector with covariance matrix $\boldsymbol{\Sigma}$ (which can be assumed to be diagonal or not), only a few adjustments are needed to calculate these two criteria (and these adjustments are the same in both cases). First, in order to integrate the noisy nature of the observations in the conditioning formulas, the distribution of $Y_N$ for any value of $N$ is obtained by replacing $C(\boldsymbol{X}, \boldsymbol{X})^{-1}$ by $(C(\boldsymbol{X}, \boldsymbol{X}) + \boldsymbol{\Sigma})^{-1}$ in equations (2) and (3). Then, as the observations are noisy, the notion of a current minimum value makes no longer sense, and we need to adapt the value of $y_{\min}$ in the proposed expressions accordingly. This value can, for example, be chosen as the value minimizing over $\mathbb{X}$ the predictive mean $\boldsymbol{x} \mapsto \mathbb{E}\left[Y_N(\boldsymbol{x})\right]$, as it is done in the knowledge gradient approaches Frazier & Powell (2007). To limit the computational cost associated with the choice of $y_{\min}$, it is also common practice to minimize the predictive mean only at the observed input points. Once these two adjustments have been made, the criteria $\text{EI}_N$ and deriv-$\text{EI}_N$ can be applied to noisy observations.

# 4 Numerical experiments

In this Section, we first illustrate the way the proposed criterion works, and the differences it implies with the classical EI criterion. In particular, it will be seen that iterates stemming from the maximization of deriv-$\text{EI}_N$ are more concentrated inside the search domain than $\text{EI}_N$ iterates. Then, by considering functions with minima inside the search domain, we show that deriv-$\text{EI}_N$ allows faster average convergence than $\text{EI}_N$ does. This is particularly visible with highly multimodal functions. In the experiments, deriv-$\text{EI}_N$ is calculated through the approximation of Eq. (15).

## 4.1 Analysis of the deriv-$\text{EI}_N$ criterion in dimension 1 and 2

**Test functions and experimental protocol**

We analyze the behavior of the deriv-$\text{EI}_N$ criterion through the study of an oscillating function in dimension $d = 1$, noted $y^{1\text{D}}$, and of a modified Branin function in dimension $d = 2$, noted $y^{2\text{D}}$ (see Figure 2 for a graphical representation of these functions, and Appendix B for their definitions). In order to focus exclusively on the effects of the acquisition criterion, we fix the hyperparameters (length scales, variance, trend parameters) of the Gaussian predictor to their maximum likelihood estimate for a large number of points. It has been observed that a good optimization of the acquisition criterion is a condition for BO to be efficient Le Riche & Picheny (2021). For this reason, the maximization of the acquisition criteria is performed by an exhaustive search on a fine grid, which is possible in such low dimensions. In higher dimension, a careful choice of the initial points of the acquisition function maximization is required Zhao et al. (2024).

**Visualizing the terms making the new acquisition criterion**

We illustrate the roles of the $\text{LikelyMin}_N$ and cond-$\text{EI}_N$ functions (which are defined in Section 3.2), by evaluating $y^{1\text{D}}$ at $N_0 = 5$ points and $y^{2\text{D}}$ at $N_0 = 12$ points randomly chosen in $\mathbb{X}$. The evolutions of $\text{LikelyMin}_N$ and cond-$\text{EI}_N$ associated to these evaluations are given in Figures 2-a and b. As expected, the function $\text{LikelyMin}_N$ is large at the points the most likely to correspond to local minima, while the function cond-$\text{EI}_N$ highlights the areas the most likely to lead to GP trajectories that have a null gradient while having values lower than the current minimum. For these particular examples, the product of the two functions, which yields the deriv-$\text{EI}_N$ criterion, favors new points inside the input domain, when the $\text{EI}_N$ criterion encourages to evaluate $y^{1\text{D}}$ (resp. $y^{2\text{D}}$) on an edge of $\mathbb{X}$. We also notice that by concentrating the search at areas of low gradient for $y^{1\text{D}}$ or $y^{2\text{D}}$, we limit the significant values of deriv-$\text{EI}_N$ to sub-regions of $\mathbb{X}$ that are smaller than what $\text{EI}_N$ would give.

**Performance of deriv-$\text{EI}_N$ over one step**

The performance of the deriv-$\text{EI}_N$ criterion is now analyzed in terms of minimization of $y^{1\text{D}}$ and $y^{2\text{D}}$.

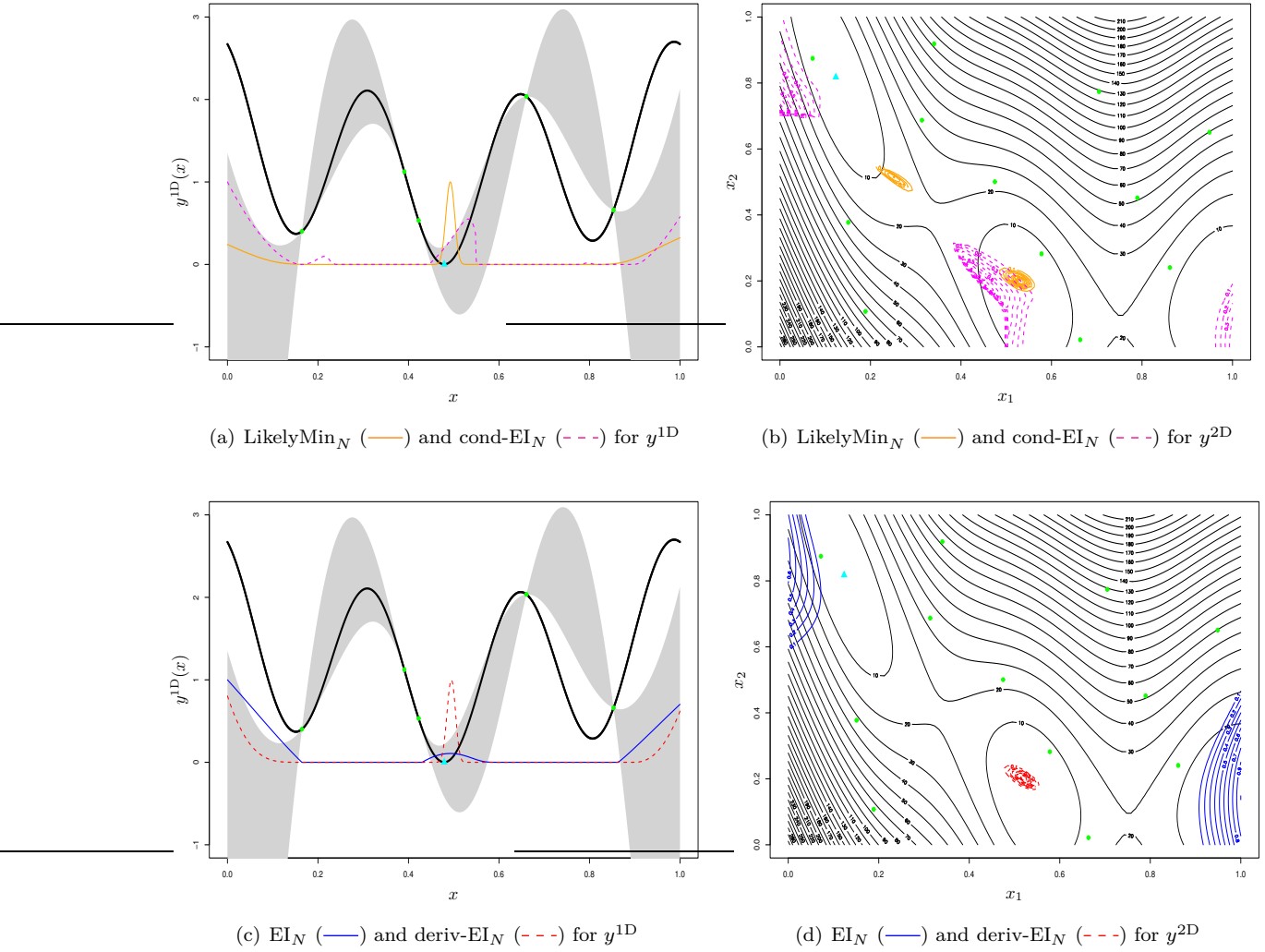

(a) LikelyMin$_N$ (——) and cond-EI$_N$ (- - -) for $y^{1\text{D}}$

(b) LikelyMin$_N$ (——) and cond-EI$_N$ (- - -) for $y^{2\text{D}}$

(c) EI$_N$ (——) and deriv-EI$_N$ (- - -) for $y^{1\text{D}}$

(d) EI$_N$ (——) and deriv-EI$_N$ (- - -) for $y^{2\text{D}}$

Figure 2: The function $y^{1\text{D}}$ is shown in black thick solid lines in plots (a) and (c) where the grey areas correspond to 95% confidence intervals of the Gaussian predictor. Identically, the black solid lines in plots (b) and (d) are the contours of function $y^{2\text{D}}$. In each plot, the global minimum is indicated by a cyan triangle, while the green dots show the points where the function has been evaluated. Plots (a) and (b) show, for the two considered functions, the evolution of $\boldsymbol{x} \mapsto \text{LikelyMin}_N(\boldsymbol{x})$ in orange solid line, and of $\boldsymbol{x} \mapsto \text{cond-EI}_N(\boldsymbol{x})$ in magenta dotted line. Plots (c) and (d) compare the evolution of $\boldsymbol{x} \mapsto \text{EI}_N(\boldsymbol{x})$ (in blue solid line) to that of $\boldsymbol{x} \mapsto \text{deriv-EI}_N(\boldsymbol{x})$ (in red dotted line). For ease of reading, the functions LikelyMin$_N$, cond-EI$_N$, EI$_N$, and deriv-EI$_N$ are normalized in such a way that their maximum value is fixed to 1.

We start with a single step. For each $j \in \{1, 2\}$, and each $k \geqslant 2d - 1$, we generate 500 space-filling LHS made of $k$ points in $\mathbb{X}$ Perrin & Cannamela (2017), which are written $\left\{ \mathcal{X}_{k,i}^{(j)} \right\}_{i=1}^{500}$. For each $1 \leqslant i \leqslant 500$, we then construct a Gaussian predictor of $y^{jD}$ based on its evaluations at each point in $\mathcal{X}_{k,i}^{(j)}$, and we denote by $\boldsymbol{x}_{k,i}^{(j),\text{deriv-EI}}$ and $\boldsymbol{x}_{k,i}^{(j),\text{EI}}$ the points of $\mathbb{X}$ maximizing the criteria deriv-EI$_k$ and EI$_k$, respectively. Let $\widehat{y}_{k,i}^{(j),\text{EI}}$ and $\widehat{y}_{k,i}^{(j),\text{deriv-EI}}$ be the smallest value of $y^{jD}$ that we obtain:

$$\widehat{y}_{k,i}^{(j),\text{EI}} := \min_{\boldsymbol{x} \in \mathcal{X}_{k,i}^{(j)} \cup \left\{ \boldsymbol{x}_{k,i}^{(j),\text{EI}} \right\}} y^{jD}(\boldsymbol{x}), \quad \widehat{y}_{k,i}^{(j),\text{deriv-EI}} := \min_{\boldsymbol{x} \in \mathcal{X}_{k}^{(i)} \cup \left\{ \boldsymbol{x}_{k,i}^{(j),\text{deriv-EI}} \right\}} y^{jD}(\boldsymbol{x}). \tag{24}$$

By construction, the lower these values are, the better the acquisition criteria should be. In this prospect, for $j \in \{1, 2\}$, Figure 3 compares the evolution of the 25%, 50% and 75% empirical quantiles of $\widehat{y}_{k,i}^{(j),\text{EI}}$ and $\widehat{y}_{k,i}^{(j),\text{deriv-EI}}$ as a function of $k$. As announced in Section 3.3, the interest of the proposed criterion lies in the intermediate (about $[5, 18] \times d$) values of $k$. For too low values, as the Gaussian predictor and its first-order derivatives are not very correlated, the criteria deriv-EI$_N$ and EI$_N$ are very close, and lead to similar results in terms of minimization of the objective function. For $k$ large, the Gaussian predictor approaches the objective function with little uncertainty, and the criteria deriv-EI$_N$ and EI$_N$ are equally capable of identifying the global minimum. For intermediate values of $k$, this phenomenon can be clearly seen in the evolution of the values of $y^{2D}$ (right plot). Because a one-dimensional space is rapidly explored, the advantage of deriv-EI$_N$ over EI$_N$ is less clear in the evolution of the values of $y^{1D}$ (left plot). There, the two criteria give almost the same results, with deriv-EI$_N$ allowing only slight improvements.

### Performance of deriv-EI$_N$ over many steps

In the above numerical experiments, one step was studied and the new evaluation points were independent of each other. Getting closer to a BO algorithm, we now quantify the effect of the acquisition criteria when defining a sequence of points where $y^{jD}$ is evaluated. To this end, for $j \in \{1, 2\}$, we generate 500 new space-filling LHS in $\mathbb{X}$ composed of 3 points each, which are written $\left\{ \widetilde{\mathcal{X}}_{3,i}^{(j)} \right\}_{i=1}^{500}$. For each $j \in \{1, 2\}$ and each repetition of the experiment $1 \leqslant i \leqslant 500$, the function $y^{jD}$ is evaluated at each point of $\widetilde{\mathcal{X}}_{3,i}^{(j)}$, and Algorithm 1 presented in Section 2 is run twice, taking as acquisition criterion deriv-EI first, then the classical criterion EI. At each iteration $k \geqslant 1$ of the algorithm, we note $y_{k,i}^{(j),\text{deriv-EI}}$ and $y_{k,i}^{(j),\text{EI}}$ the obtained current minima of $y^{jD}$. The empirical estimates of the median and the mean of these current minima is shown in Figure 4. The interest of the deriv-EI acquisition criterion is again underlined by these results. Indeed, for all iterations $k$, the median and the mean of the current minima associated with the deriv-EI criterion are lower than those of the current minima associated with the EI criterion. Again, the advantage of deriv-EI over EI takes place in the middle of the iterations $k$. Note that the median is well below the mean for the minimization of $y^{1D}$. It comes from the fact that, for both EI and deriv-EI, some of the runs have taken a significant number of iterations to identify the area of the global minimum.

### 4.2 Performance analysis in dimensions 2, 3 and 5

### Test functions construction

The EI and deriv-EI acquisition criteria are now compared on a larger set of test functions. To define this set of functions, we elaborate on the idea of using GPs Hennig & Schuler (2012) which are by construction compatible with the working assumptions. We start by noting $Z_\theta^{(d)}$ the Gaussian process defined on $\mathbb{X} = [0, 1]^d$ such that for any $\boldsymbol{x}, \boldsymbol{x}' \in \mathbb{X}$ and any $\theta > 0$

$$\mathbb{E}\left[ Z_\theta^{(d)}(\boldsymbol{x}) \right] = 0, \quad \text{Cov}(Z_\theta^{(d)}(\boldsymbol{x}), Z_\theta^{(d)}(\boldsymbol{x}')) = \prod_{i=1}^{d} \kappa \left( \sqrt{\frac{2}{d}} \frac{|x_i - x_i'|}{\theta} \right), \tag{25}$$

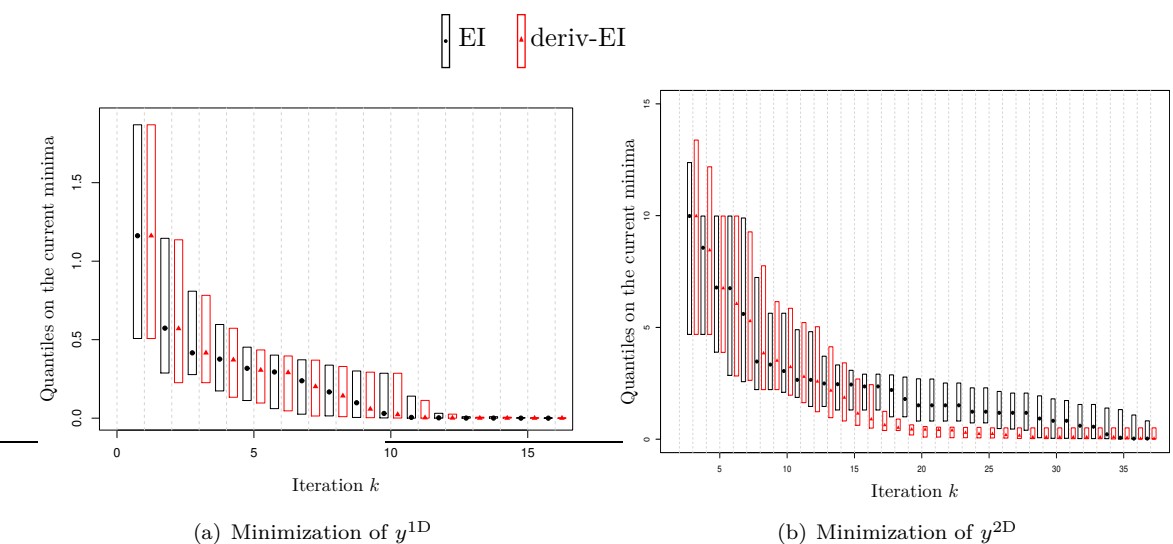

(a) Minimization of $y^{1D}$

(b) Minimization of $y^{2D}$

Figure 3: Influence of the acquisition criterion deriv-EI$_N$ and EI$_N$ when minimizing $y^{1D}$ and $y^{2D}$. For $k \geqslant 2d - 1$, the lower and upper parts of the black rectangles correspond to the 25% and 75% quantiles of $\widehat{y}_{k,i}^{(j),\mathrm{EI}}$, while the black circles show the median value. Similarly, the lower and upper parts of the red rectangles correspond to the 25% and 75% quantiles of $\widehat{y}_{k,i}^{(j),\mathrm{deriv\text{-}EI}}$, while the red triangles show the median value.

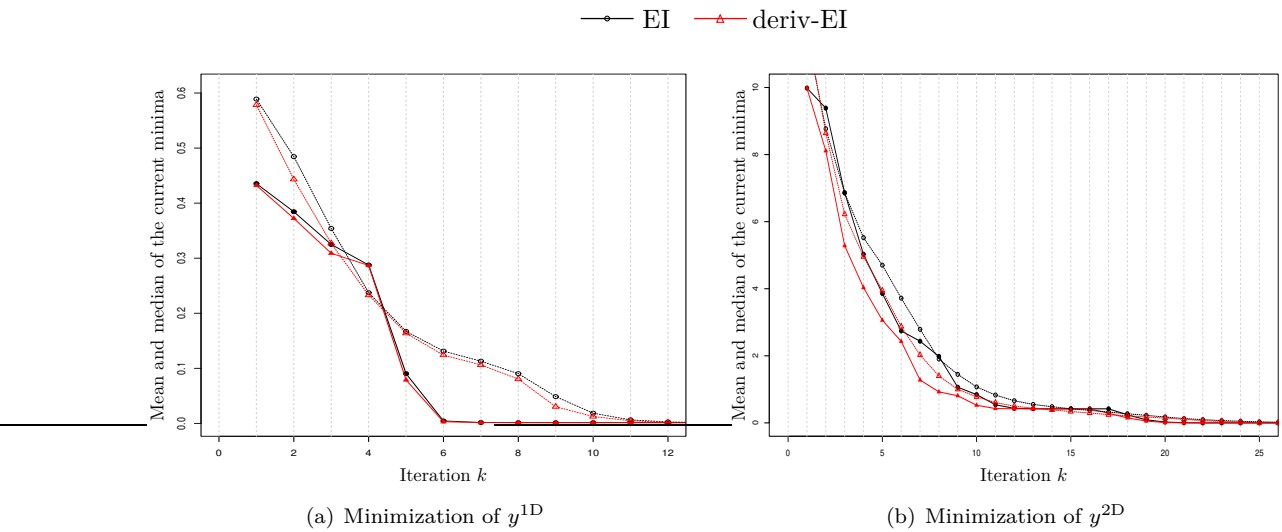

(a) Minimization of $y^{1D}$

(b) Minimization of $y^{2D}$

Figure 4: Influence of the acquisition criterion deriv-EI$_N$ and EI$_N$ when minimizing $y^{1D}$ and $y^{2D}$. For $k \geqslant 1$, the filled black circles (resp. the filled red triangles) represent the empirical median of $\left\{ y_{k,i}^{(j),\mathrm{EI}} \right\}_{i=1}^{500}$ (resp. of $\left\{ y_{k,i}^{(j),\mathrm{deriv\text{-}EI}} \right\}_{i=1}^{500}$), and the empty black circles (resp. empty red triangles) indicate the empirical means.

where $\kappa$ is the Matérn-5/2 covariance function of Equation (9), which is such that $Z_\theta^{(d)}$ is twice differentiable in the mean-square sense. Notice the normalization of the length scales in $\sqrt{d/2}$, allowing to define Gaussian processes in any dimension $d$ with close dependence structures. This normalization can also be understood by seeing that distances (between the two farthest points, or the expected distance of two points randomly drawn in $\mathbb{X}$) grow in $\sqrt{d}$, therefore the length scales have to grow in $\sqrt{d}$. We consider as test function class the set $\mathcal{F}_\theta^{(d)}$ of realizations of $Z_\theta^{(d)}$ that admits a global minimum strictly inside $\mathbb{X}$ (i.e., at a point of zero partial derivatives). The following numerical tests then focus on two particular values of $\theta$: $\theta = 0.2$ will characterize strongly oscillating functions admitting a large number of local minima, while $\theta = 0.5$ will refer to more regular functions presenting a smaller number of local minima. For $\theta \in \{0.2, 0.5\}$ and $d \in \{2, 3, 5\}$, we generate 100 functions from $\mathcal{F}_\theta^{(d)}$ in a random and independent way. These functions are noted $\left\{ y_{i,\theta}^{(d)} \right\}_{i=1}^{100}$ (see Appendix C for a detailed description of their construction). We finally subtract from each function its minimum value so that

$$\min_{\boldsymbol{x} \in \mathbb{X}} y_{i,\theta}^{(d)}(\boldsymbol{x}) = 0, \tag{26}$$

and we proceed to the same shifting on the $Y$ process. Figure 5 shows four examples of such functions belonging to $\mathcal{F}_{0.2}^{(2)}$ and $\mathcal{F}_{0.5}^{(2)}$ in the case $d = 2$.

**Experimental protocol**

The global minimum of these functions is then searched twice with Algorithm 1 by, first, taking deriv-EI and, then, EI as the acquisition criterion. The total number of calls to the objective function of each optimization run is equal to budget = 100. The two types of searches are initialized with the evaluation of $y_{i,\theta}^{(d)}$ at the same space-filling LHS of dimension $N_0 = 3$ (a different design is generated for each function minimization). The size of the design is small and does not depend on $d$. As observed in Le Riche & Picheny (2021); Hutter et al. (2013), small random designs at the beginning of BO searches are more efficient. Moreover, the effect of the acquisition criterion is more visible for small initial random designs. The growth of the length scales in $\sqrt{d}$ (Equation 25) guarantees that the correlation between the $N_0$ points is the same, independently of $d$. In order to investigate the influence of the acquisition criterion only on the optimization but not on the learning of the GP, the properties of the Gaussian process $Y$ used to guide the search are chosen equal to those of $Z_\theta^{(d)}$. The maximization of the acquisition criteria is performed in two steps: each acquisition criterion is first evaluated in $10^5$ points randomly chosen in $\mathbb{X}$, and 10 Nelder-Mead algorithms starting from the 10 most promising points among the random points are then launched in parallel to identify the new point at which to evaluate the objective function.

Two quantities of interest are then extracted from these Bayesian optimizations. For each $1 \leqslant k \leqslant$ budget, each $d \in \{2, 3, 5\}$, and each $\theta \in \{0.2, 0.5\}$, we first note $\widehat{y}_\theta^{(d),\text{deriv-EI}}(k)$ (resp. $\widehat{y}_\theta^{(d),\text{EI}}(k)$) the empirical mean of the current minimum (mean best-so-far performance) obtained at the $k^{\text{th}}$ iteration on all the tested functions when taking deriv-EI (resp. EI) as the acquisition criterion. Second, we define $\widehat{k}^{(j),\text{deriv-EI}}(\theta, s)$ (resp. $\widehat{k}^{(j),\text{EI}}(\theta, s)$), the mean time-to-target that is the average number of iterations necessary for the best-so-far observation to be lower than a threshold $s > 0$ when using deriv-EI (resp. EI). Note that for both quantities of interest, the average is done on the different test-functions, which have the same kind of variations and the same minimum equal to 0, which makes them comparable although potentially very different.

**Optimization results**

The evolution of these quantities of interest are shown in Figure 6 for $\theta = 0.2$ and Figure 7 for $\theta = 0.5$. In all of these figures, a substantial gain is brought by the deriv-EI criterion with respect to the EI criterion. The gain is visible both in terms of the mean best-so-far objective function (plots (a) to (c)) and the mean time-to-target (plots (d) to (f)). We notice, as we had hoped in Section 3.3, that the observed improvements

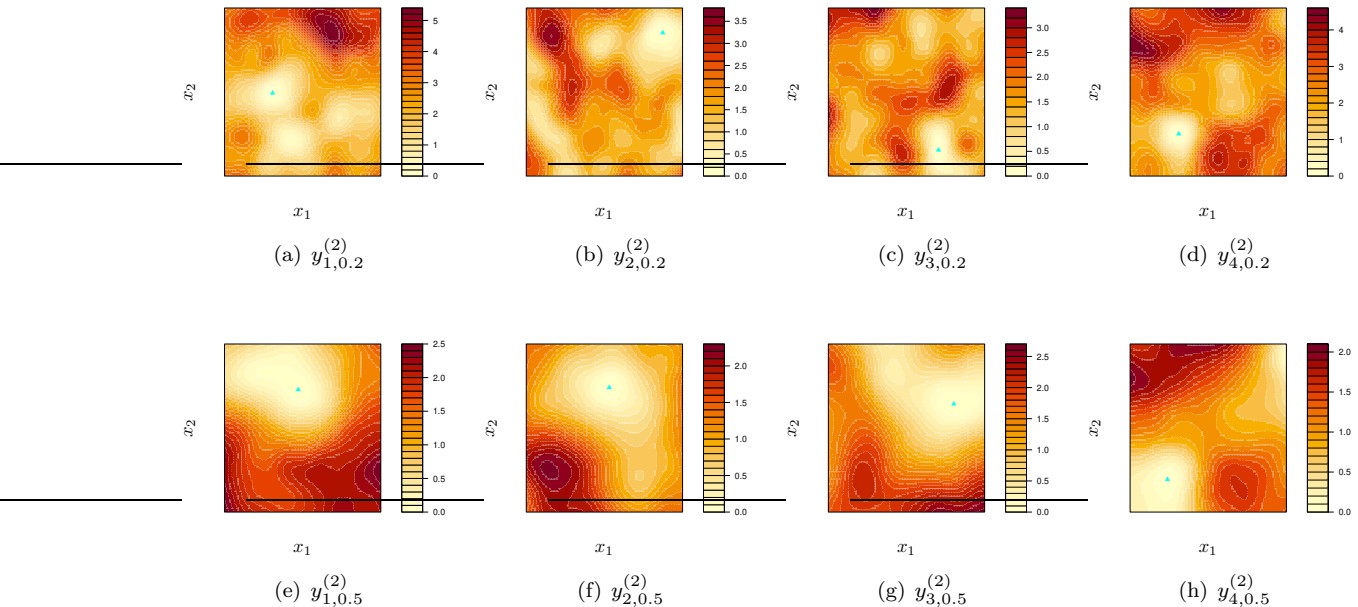

Figure 5: Representation of 4 particular elements of $\mathcal{F}_{0.2}^{(2)}$ and $\mathcal{F}_{0.5}^{(2)}$, the set of test functions to be minimized. In each function instance, the global minimum location is indicated with a cyan triangle.

brought by the deriv-EI criterion are greater as the dimension $d$ of the input space increases. As expected, we also observe that choosing deriv-EI rather than EI is of more interest for more multimodal functions, i.e., when the length scale $\theta$ is small. Indeed, it is in these configurations with a large number of local minima that adding information about null first order derivatives and positive definite Hessian matrices is useful.

**Remark** For all the test functions studied in this section, the condition numbers of the covariance matrices of the observation points were all between $10^3$ and $2 \times 10^6$.

## 5 Extensions and conclusions

### 5.1 Management of minima on bounds

The article has assumed until now that the minimum is inside the search space $\mathbb{X}$. If this is not the case, orienting the search towards areas with a zero gradient can actually be counterproductive, as the global minimum will typically be associated with a nonzero gradient. Denoting by $\partial \mathbb{X}$ the boundary of $\mathbb{X}$, a first possibility to circumvent this problem is to penalize the objective function so that optima on the boundary are pushed inside the domain but arbitrarily close to the boundary, and therefore are associated to a null gradient and positive definite Hessian. This is the idea of the barrier functions of the interior point methods Wright & Nocedal (2006). With barrier functions, the objective $\boldsymbol{x} \mapsto y(\boldsymbol{x})$ is replaced by $\boldsymbol{x} \mapsto y(\boldsymbol{x}) + \lambda c(\boldsymbol{x})$, where $\lambda$ is a positive constant and $c$ is a continuously twice-differentiable positive function such that $c(\boldsymbol{x})$ would be close to zero when $\boldsymbol{x}$ is far from the boundaries of $\mathbb{X}$, and would take potentially infinite values when $\boldsymbol{x} \in \partial \mathbb{X}$. For instance, if $\mathbb{X} = [0, 1]$, the function $c$ can be chosen as:

$$c(x) = \frac{1}{\min(x, 1-x)} \quad \text{or} \quad c(x) = -\log(\min(x, 1-x)) \quad , \quad 0 \leqslant x \leqslant 1. \tag{27}$$

The larger $\lambda$ is, the further from $\partial \mathbb{X}$ the global minimum of $y + \lambda c$ is, whether the global minimum of $y$ is on $\partial \mathbb{X}$ or not. And by making $\lambda$ progressively tend towards 0, we make this global minimum, which

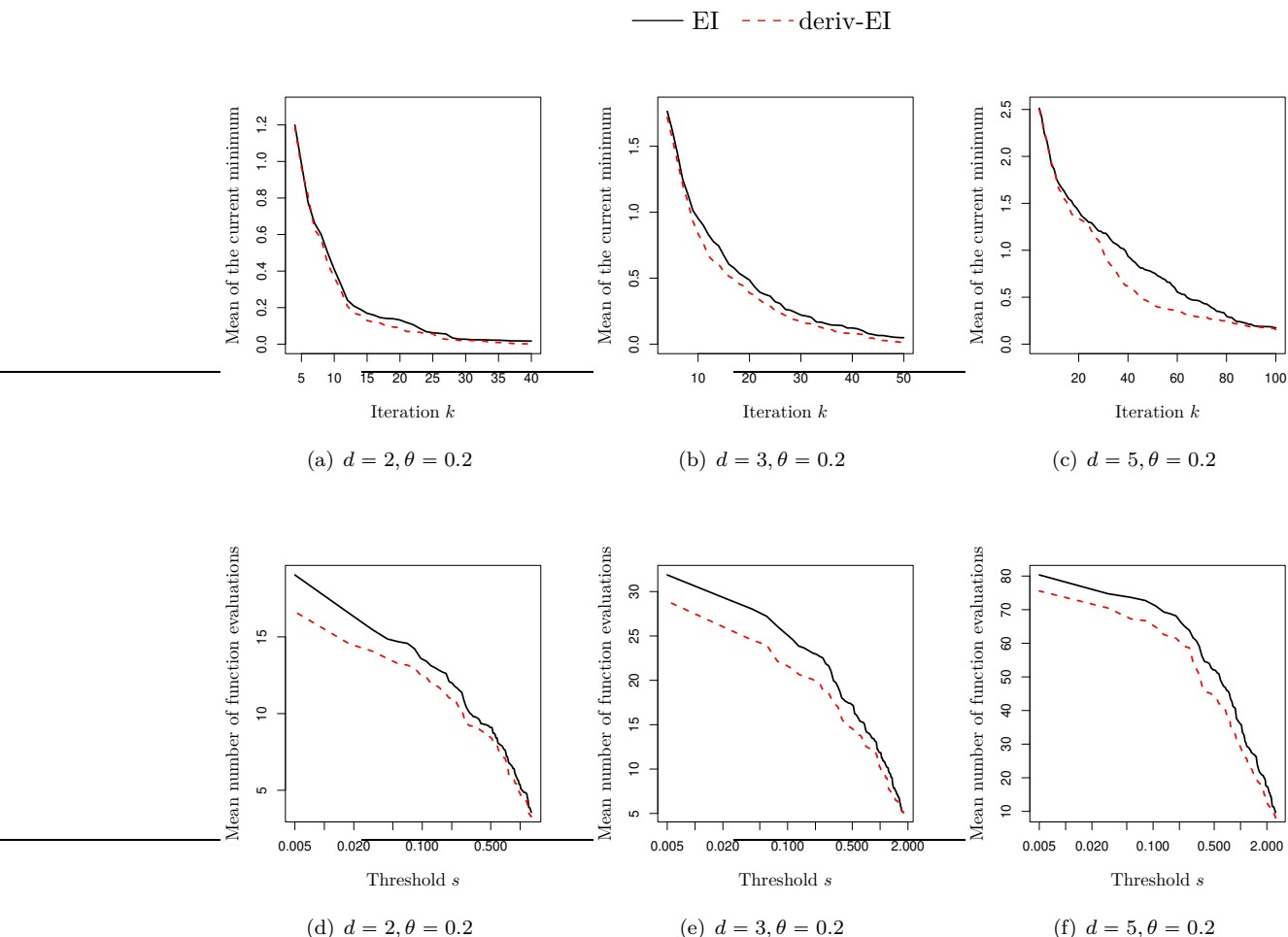

Figure 6: Plots (a), (b), and (c) show the mean best performance, $k \mapsto \widehat{y}_{\theta}^{(d),\text{EI}}(k)$ (in black solid line ——) and $k \mapsto \widehat{y}_{\theta}^{(d),\text{deriv-EI}}(k)$ (in red dotted line - - -) for strongly multimodal functions ($\theta = 0.2$) and $d \in \{2, 3, 5\}$. Plots (d), (e), and (f) give the mean time-to-target $s \mapsto \widehat{k}^{(j),\text{EI}}(\theta, s)$ (in black solid line) and $s \mapsto \widehat{k}^{(j),\text{deriv-EI}}(\theta, s)$ (in red dotted line).

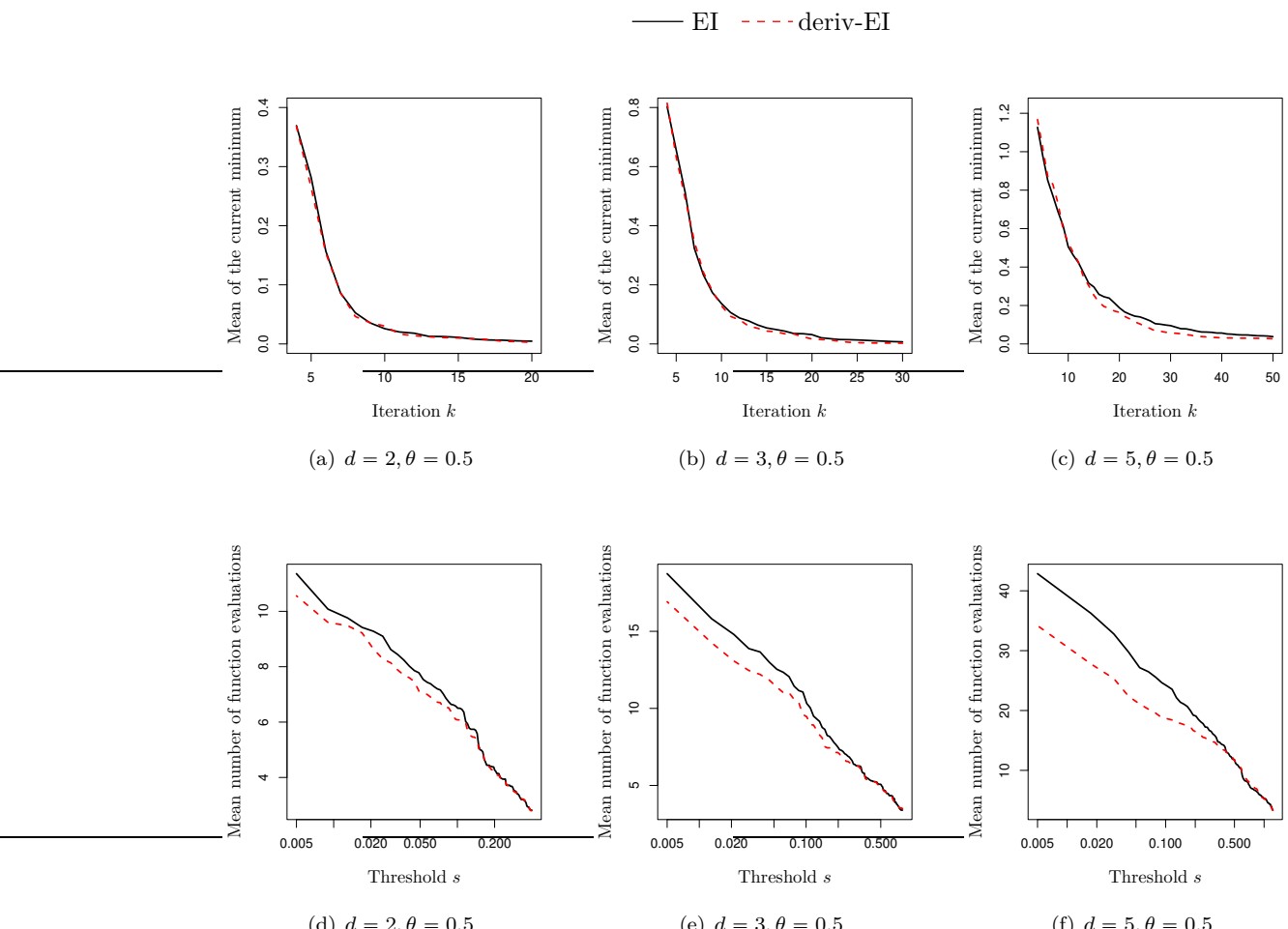

Figure 7: Plots (a), (b), and (c) compare the mean best-so-far performances for the two acquisition criteria, $k \mapsto \widehat{y}_{\theta}^{(d),\text{EI}}(k)$ (in black solid line ——) and $k \mapsto \widehat{y}_{\theta}^{(d),\text{deriv-EI}}(k)$ (in red dotted line - - -), for moderatly multimodal functions ($\theta = 0.5$) and and $d \in \{2, 3, 5\}$. Plots (d), (e), and (f) compare the mean time-to-target, $s \mapsto \widehat{k}^{(j),\text{EI}}(\theta, s)$ (in black solid line) and $s \mapsto \widehat{k}^{(j),\text{deriv-EI}}(\theta, s)$ (in red dotted line).

will be well associated to a zero gradient of $y + \lambda c$, tend to the global minimum of $y$. Such an approach is well-studied for convex optimization problems, with bounds linking the choice of $\lambda$ to the degradation in the optimal value of the objective function Wright & Nocedal (2006). In general however, the effect of the $\lambda$ decay law is difficult to understand. And the penalty adds a steep function increase at the edge of $\mathbb{X}$ that GPR-based metamodels will have difficulty to learn.

Alternatively, we propose to evaluate *a priori* the likelihood, noted $\ell_N$, that the minimum of $y$ lies on the boundary of $\mathbb{X}$.

The $\ell_N$ likelihood can be evaluated through a three-step procedure. Starting from the GP-based surrogate model of $y$, noted $Y_N$, the first step would be to look for positions that may correspond to local minima of $y$, by running in parallel $M \gg 1$ regularized Newton descent algorithms on the trajectories of $Y_N$. These minimizations would be in that case initialized at randomly chosen points $\boldsymbol{x}_0^m \in \mathbb{X}$, $1 \leqslant m \leqslant M$, and we would denote by $\boldsymbol{x}_*^m$ the obtained minimum when starting from $\boldsymbol{x}_0^m$. Then, we generate $Q \gg 1$ random samples of the Gaussian random vector $(Y_N(\boldsymbol{x}_*^1), \ldots, Y_N(\boldsymbol{x}_*^M))$, which we denote by

$$(Y_N^{\omega_q}(\boldsymbol{x}_*^1), \ldots, Y_N^{\omega_q}(\boldsymbol{x}_*^M)), \; 1 \leqslant q \leqslant Q. \tag{28}$$

The indicator $\ell_N$ is finally estimated by counting how often the minima of the draws are on the bounds,

$$\ell_N := \frac{1}{Q} \sum_{q=1}^{Q} 1_{\boldsymbol{x}_*^{m_q^\star} \in \partial \mathbb{X}} \quad , \quad \boldsymbol{x}_*^{m_q^\star} \in \arg \min_{1 \leqslant m \leqslant M} Y_N^{\omega_q}(\boldsymbol{x}_*^m). \tag{29}$$

Depending on the value of $\ell_N$, the method described in the rest of the article can be complemented in one of the two following fashions. If $\ell_N$ is too large, the traditional $\mathrm{EI}_N$ acquisition criterion should be used instead of deriv-$\mathrm{EI}_N$. Alternatively, $\ell_N$ can be calculated specifically for each bound and if it is likely that some specific bounds are hit, then the corresponding variables can be set to these bounds, the BO iteration being carried out with the deriv-$\mathrm{EI}_N$ criterion in the lower dimensional space.

Nonetheless, the objective of this work was to come up with an acquisition criterion applicable when the minimum of $y$ is not on the boundary of $\mathbb{X}$. We leave the continuation of the above analysis, based on the likelihood to have the optimum at a bound, as a perspective to this work.

## 5.2 Summary and further perspectives

In the context of Bayesian optimization, this work proposes a novel acquisition criterion allowing to integrate as additional *a priori* the fact that interior minima are associated to zero first order derivatives and positive definite Hessians. With this addition, a classical acquisition criterion such as the expected improvement takes on a feature of information theoretic criteria by characterizing the distribution of potential optima when the plain expected improvement accounts for all improving values of the function model. The new expected improvement with derivatives, called deriv-EI, does not need the derivatives of the true function. A computationally efficient approximation to deriv-EI is proposed in the article.

It has been observed through several test cases that the new criterion allows significant gains in terms of function minimization at intermediate budgets of function evaluations. This benefit is larger when the function to minimize presents several local minima or the dimension is high, since in these cases the classical expected improvement is too exploratory in particular in areas near the bounds Siivola et al. (2018).

All the consequences of the proposed acquisition criterion could not be investigated in this paper. In order to simplify the interpretation of the results, all the comparisons between the classical EI and the deriv-EI criteria have been carried out in *ideal* configurations in the sense that the test functions are realizations of the Gaussian process guiding the minimization. The hyperparameters of the GP characterizing its

statistical properties are always known by construction of the test functions. Therefore, it will be interesting to study the sensitivity of Bayesian optimization with deriv-EI to the iterative estimation of the GP hyperparameters, as it happens in practice. In the same manner, only problems in moderate dimensions are implemented ($d \leqslant 5$), as it seems important to restrict ourselves to cases for which the maximization of the EI and deriv-EI criteria can be sufficiently well solved. During the analysis of configurations in higher dimensions ($d \geqslant 10$), the maximization of these criteria becomes a problem in itself and the performances of the EI and deriv-EI criteria turned out to be too dependent on our ability to correctly maximize them. Working at the definition of efficient methods to maximize the deriv-EI criterion would therefore be an appropriate continuation to this work.

Finally, this work has focused exclusively on the EI acquisition criterion because it is a standard in BO, but other acquisition criteria should also benefit from the predictor's derivatives. For example, the EI criterion of Equation (12) can be adapted to the related probability of improvement Kushner (1964): instead of maximizing $\boldsymbol{x} \mapsto \mathbb{P}(Y_N(\boldsymbol{x}) < y_{\min}) = \mathbb{E}\left[1_{Y_N(\boldsymbol{x}) < y_{\min}}\right]$, we could maximize

$$\boldsymbol{x} \mapsto \mathbb{E}\left[1_{Y_N(\boldsymbol{x}) < y_{\min}} \times 1_{\boldsymbol{R}_N(\boldsymbol{x})\partial Y_N(\boldsymbol{x}) \in \mathcal{B}(\varepsilon), \partial^2 Y_N(\boldsymbol{x}) \in \mathcal{M}^+(d)}\right]. \tag{30}$$

In the same manner, the well-known upper confidence bound acquisition criterion Srinivas et al. (2009) could be adapted by adding a penalty term on the derivatives, which would make it possible to explicitly play on the exploitation vs. exploration tradeoff in the same way as the standard deviation. The new point to be evaluated could for instance be sought as a solution to the following problem:

$$\boldsymbol{x}^{(N+1)} \in \underset{\boldsymbol{x} \in \mathbb{X}}{\arg\min}\, \mu_N(\boldsymbol{x}) - \lambda_1 \sigma_N(\boldsymbol{x}) + \lambda_2 \left\|\mathbb{E}\left[\partial Y_N(\boldsymbol{x})\right]\right\|, \tag{31}$$

with $\lambda_1$ and $\lambda_2$ two positive constants.

### Acknowledgments

This action benefited from the support of the Chair Stress Test, Risk Management and Financial Steering, led by the French Ecole polytechnique and its foundation and sponsored by BNP Paribas. This research also benefited from the consortium in Applied Mathematics CIROQUO (https://doi.org/10.5281/zenodo.6581217), gathering partners in technological and academia in the development of advanced methods for Computer Experiments.

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

## A  Approximation of the EI with derivatives information

We start by recalling the notations used in Equations (19) and (20). For each $\boldsymbol{x} \in \mathbb{X}$, the means and covariance matrices of random vectors $\partial Y_N(\boldsymbol{x})$ and $(Y_N(\boldsymbol{x}), D^2 Y_N(\boldsymbol{x}))|\partial Y_N(\boldsymbol{x}) = \boldsymbol{0}$ are written

$$\partial Y_N(\boldsymbol{x}) \ \sim \ \mathcal{N}\left(\dot{\boldsymbol{m}}, \dot{\boldsymbol{S}}\right),$$

$$(Y_N(\boldsymbol{x}), D^2 Y_N(\boldsymbol{x}))|\partial Y_N(\boldsymbol{x}) = \boldsymbol{0} \ \sim \ \mathcal{N}\left( \begin{pmatrix} m \\ \ddot{m}_1 \\ \vdots \\ \ddot{m}_d \end{pmatrix}, \begin{bmatrix} s^2 & \rho_{1,1} & \cdots & \rho_{1,d} \\ \rho_{1,1} & \ddot{s}_1 & \ddots & \vdots \\ \vdots & \ddots & \ddots & \rho_{d-1,d} \\ \rho_{d,1} & \cdots & \rho_{d,d-1} & \ddot{s}_d \end{bmatrix} \right),$$

where the notation $D^2 Y_N := \left((\partial^2 Y_N)_{1,1}, \ldots, (\partial^2 Y_N)_{d,d}\right)$ refers to the diagonal terms of the matrix $\partial^2 Y_N$.

Using the notations and the general expressions introduced in Section 3.1, the expressions of $m$ and $s^2$ come from the conditioning of the $(d+1)$-dimensional Gaussian vector $(Y_N(\boldsymbol{x}), \partial Y_N(\boldsymbol{x}))$, whose mean is $(\mu_N(\boldsymbol{x}), \partial\mu_N(\boldsymbol{x}))$, and whose covariance matrix is

$$\begin{bmatrix} C_N(\boldsymbol{x}, \boldsymbol{x}) & \partial C_N(\boldsymbol{x}, \boldsymbol{x})^T \\ \partial C_N(\boldsymbol{x}, \boldsymbol{x}) & \partial^2 C_N(\boldsymbol{x}, \boldsymbol{x}) \end{bmatrix}.$$

Applying the conditioning formula yields,

$$m := \mathbb{E}[Y_N(\boldsymbol{x})|\partial Y_N(\boldsymbol{x}) = \boldsymbol{0}] = \mu_N(\boldsymbol{x}) + \partial C_N(\boldsymbol{x}, \boldsymbol{x})^\top \partial^2 C_N(\boldsymbol{x}, \boldsymbol{x})^{-1}(\boldsymbol{0} - \partial\mu_N(\boldsymbol{x})) \ ,$$

$$s^2 := \text{Var}(Y_N(\boldsymbol{x})|\partial Y_N(\boldsymbol{x}) = \boldsymbol{0}) = C_N(\boldsymbol{x}, \boldsymbol{x}) - \partial C_N(\boldsymbol{x}, \boldsymbol{x})^\top \partial^2 C_N(\boldsymbol{x}, \boldsymbol{x})^{-1} \partial C_N(\boldsymbol{x}, \boldsymbol{x}) \ .$$

The expressions for the mean and covariance matrix of the Gaussian vector $Y_N(\boldsymbol{x}), D^2 Y_N(\boldsymbol{x})|\partial Y_N(\boldsymbol{x}) = \boldsymbol{0}$, i.e., the symbols $\ddot{m}_i$, $\ddot{s}_i$ and $\rho_{i,j}$ in Equation (19), are obtained following the same conditioning principle, but

applied to the Gaussian vector $(Y_N(\boldsymbol{x}), D^2Y_N(\boldsymbol{x}), \partial Y_N(\boldsymbol{x}))$, whose mean is $(\mu_N(\boldsymbol{x}), D^2\mu_N(\boldsymbol{x}), \partial\mu_N(\boldsymbol{x}))$, and whose covariance matrix is made of properly ordered terms such as $\mathrm{Cov}\left(\frac{\partial^2 Y_N}{\partial x_i^2}(\boldsymbol{x}), \frac{\partial Y_N}{\partial x_j}(\boldsymbol{x})\right) = \frac{\partial^3 C_N}{\partial x_i^2 \partial x_j}(\boldsymbol{x}, \boldsymbol{x})$, $\mathrm{Cov}\left(\frac{\partial^2 Y_N}{\partial x_i^2}(\boldsymbol{x}), \frac{\partial^2 Y_N}{\partial x_j}(\boldsymbol{x})\right) = \frac{\partial^4 C}{\partial x_i^2 \partial^2 x_j}(\boldsymbol{x}, \boldsymbol{x})$, ....

The conditioning can be applied one more time to $(Y_N(\boldsymbol{x}), D^2Y_N(\boldsymbol{x})|\partial Y_N(\boldsymbol{x}) = \boldsymbol{0})$ to account for an observation of $Y_N(\boldsymbol{x})$, leading to

$$(D^2Y_N(\boldsymbol{x}))_i | \partial Y_N(\boldsymbol{x}) = \boldsymbol{0}, Y_N(\boldsymbol{x}) = y \sim \mathcal{N}(\ddot{m}_i + \rho_{1i}(y-m)/s^2, \ddot{s}_i^2 - \rho_{1i}^2/s^2). \tag{32}$$

For $p \in \{1, 2\}$, let us first assume that the off-diagonal terms of $\partial^2 \ddot{Y}_N$ can be neglected. In that case, ensuring that $\partial^2 Y_N$ is in $\mathcal{M}^+(d)$ comes down to ensuring that its diagonal terms are positive, i.e. ensuring that $D^2Y_N$ is in $[0, +\infty[^d$, and we can write (using the former notations):

$$\begin{aligned}
\mathrm{deriv\text{-}EI}_N(\boldsymbol{x}) &:= \int_{y=-\infty}^{y_{\min}} \int_{\dot{\boldsymbol{y}} \in \mathcal{E}(\boldsymbol{x}, \varepsilon)} \int_{\ddot{\boldsymbol{Y}} \in \mathcal{M}^+(d)} (y_{\min} - y)^p d\mathbb{P}(y, \dot{\boldsymbol{y}}, \ddot{\boldsymbol{Y}}) \\
&\approx \int_{y=-\infty}^{y_{\min}} \int_{\dot{\boldsymbol{y}} \in \mathcal{E}(\boldsymbol{x}, \varepsilon)} \int_{\ddot{\boldsymbol{y}} \in [0, +\infty[^d} (y_{\min} - y)^p f_{\partial Y_N(\boldsymbol{x})}(\dot{\boldsymbol{y}}) f_{Y_N(\boldsymbol{x}), D^2Y_N(\boldsymbol{x})|\partial Y_N(\boldsymbol{x})=\dot{\boldsymbol{y}}}(y, \ddot{\boldsymbol{y}}) dy d\dot{\boldsymbol{y}} d\ddot{\boldsymbol{y}}.
\end{aligned} \tag{33}$$

As it is necessary for a matrix to have positive terms on its diagonal to be positive definite, this approximation is an overestimation of the number of trajectories that are actually positive definite. In addition, if $\varepsilon$, the size of the ellipsoid centered at $\boldsymbol{0}$ to which $\partial Y_N(\boldsymbol{x})$ belongs, is sufficiently small, it is possible to approximate $f_{\partial Y_N(\boldsymbol{x})}(\dot{\boldsymbol{y}})$ by $f_{\partial Y_N(\boldsymbol{x})}(\boldsymbol{0})$ for any $\dot{\boldsymbol{y}}$ in $\mathcal{E}(\boldsymbol{x}, \varepsilon)$, which leads to:

$$\begin{aligned}
&\mathrm{deriv\text{-}EI}_N(\boldsymbol{x}) \\
&\approx \mathrm{Vol}(\mathcal{E}(\boldsymbol{x}, \varepsilon)) f_{\partial Y_N(\boldsymbol{x})}(\boldsymbol{0}) \int_{y=-\infty}^{y_{\min}} \int_{\ddot{\boldsymbol{y}} \in [0, +\infty[^d} (y_{\min} - y)^p f_{Y_N(\boldsymbol{x}), D^2Y_N(\boldsymbol{x})|\partial Y_N(\boldsymbol{x})=\boldsymbol{0}}(y, \ddot{\boldsymbol{y}}) dy d\ddot{\boldsymbol{y}}.
\end{aligned} \tag{34}$$

where $\mathrm{Vol}(\mathcal{E}(\boldsymbol{x}, \varepsilon))$ is the volume of $\mathcal{E}(\boldsymbol{x}, \varepsilon)$.

We further assume that for any $y \in \mathbb{R}$, the components of $D^2Y_N(\boldsymbol{x})$ conditioned by the event $(\partial Y_N(\boldsymbol{x}) = \boldsymbol{0}, Y_N(\boldsymbol{x}) = y)$ are statistically independent. In other words, a trajectory which passes through $(\boldsymbol{x}, y)$ and which is flat is assumed to have independent curvatures. In this case, the density of $D^2Y_N(\boldsymbol{x})|(\partial Y_N(\boldsymbol{x}) = \boldsymbol{0}, Y_N(\boldsymbol{x}) = y)$ is a product of univariate densities. This leads to,

$$\begin{aligned}
&\int_{y=-\infty}^{y_{\min}} \int_{\ddot{\boldsymbol{y}} \in [0, +\infty[^d} (y_{\min} - y)^p f_{Y_N(\boldsymbol{x}), D^2Y_N(\boldsymbol{x})|\partial Y_N(\boldsymbol{x})=\boldsymbol{0}}(y, \ddot{\boldsymbol{y}}) dy d\ddot{\boldsymbol{y}} \\
&= \int_{y=-\infty}^{y_{\min}} \int_{\ddot{\boldsymbol{y}} \in [0, +\infty[^d} (y_{\min} - y)^p f_{Y_N(\boldsymbol{x})|\partial Y_N(\boldsymbol{x})=\boldsymbol{0}}(y) f_{D^2Y_N(\boldsymbol{x})|\partial Y_N(\boldsymbol{x})=\boldsymbol{0}, Y_N(\boldsymbol{x})=y}(\ddot{\boldsymbol{y}}) dy d\ddot{\boldsymbol{y}} \\
&\approx \int_{y=-\infty}^{y_{\min}} \frac{(y_{\min} - y)^p}{(2\pi)^{\frac{d+1}{2}}} \exp\left(-\frac{(y-m)^2}{2s^2}\right) \left(\prod_{i=1}^{d} \int_{\ddot{y}_i=0}^{+\infty} \exp\left(-\frac{(\ddot{y}_i - (\ddot{m}_i + \rho_{1i}(y-m)/s^2))^2}{2\ddot{s}_i^2(1 - \rho_{1i}^2/(s^2\ddot{s}_i^2))}\right) \frac{d\ddot{y}_i}{\ddot{s}_i\sqrt{1 - \frac{\rho_{1i}}{s\ddot{s}_i}}}\right) \frac{dy}{s}.
\end{aligned} \tag{35}$$

If we now perform the following variable changes: $z := (y-m)/s$, $z_{\min} := (y_{\min} - m)/s$, $r_i := \rho_{1i}/(s\ddot{s}_i)$ and $\ddot{\tau}_i = \ddot{m}_i/\ddot{s}_i$, it comes:

$$\int_{y=-\infty}^{y_{\min}} \frac{(y_{\min}-y)^p}{(2\pi)^{\frac{d+1}{2}}} \exp\left(-\frac{(y-m)^2}{2s^2}\right) \left(\prod_{i=1}^{d} \int_{\ddot{y}_i=0}^{+\infty} \exp\left(-\frac{(\ddot{y}_i - (\ddot{m}_i + \rho_{1i}(y-m)/s^2))^2}{2\ddot{s}_i^2(1-\rho_{1i}^2/(s^2\ddot{s}_i^2))}\right) \frac{d\ddot{y}_i}{\ddot{s}_i\sqrt{1-\frac{\rho_{1i}}{s\ddot{s}_i}}}\right) \frac{dy}{s}$$

$$= s^p \int_{z=-\infty}^{z_{\min}} \frac{(z_{\min}-z)^p}{(2\pi)^{\frac{d+1}{2}}} \exp\left(-\frac{z^2}{2}\right) \left(\prod_{i=1}^{d} \int_{\ddot{z}_i=-\ddot{\tau}_i}^{+\infty} \exp\left(-\frac{(\ddot{z}_i - r_i z)^2}{2(1-r_i^2)}\right) \frac{d\ddot{z}_i}{\sqrt{1-r_i^2}}\right) dz$$

$$= s^p \int_{z=-\infty}^{z_{\min}} \frac{(z_{\min}-z)^p}{\sqrt{2\pi}} \exp\left(-\frac{z^2}{2}\right) \prod_{i=1}^{d} \Phi\left(\frac{\ddot{\tau}_i + r_i z}{\sqrt{1-r_i^2}}\right) dz. \tag{36}$$

Recalling that $\Phi$ and $\phi$ are respectively the CDF and the PDF of the standard Gaussian variables, the former expression can be further simplified by introducing the following first order Taylor expansion of the function $\Phi$,

$$\Phi\left(\frac{\ddot{\tau}_i + r_i z}{\sqrt{1-r_i^2}}\right) \approx \Phi\left(\frac{\ddot{\tau}_i}{\sqrt{1-r_i^2}}\right) + \frac{r_i z}{\sqrt{1-r_i^2}}\phi\left(\frac{\ddot{\tau}_i}{\sqrt{1-r_i^2}}\right), \tag{37}$$

and by truncating to the first polynomial orders, so that:

$$\int_{z=-\infty}^{z_{\min}} \frac{(z_{\min}-z)^p}{\sqrt{2\pi}} \exp\left(-\frac{z^2}{2}\right) \prod_{i=1}^{d} \Phi\left(\frac{\ddot{\tau}_i + r_i z}{\sqrt{1-r_i^2}}\right) dz$$

$$\approx \int_{z=-\infty}^{z_{\min}} \frac{(z_{\min}-z)^p}{\sqrt{2\pi}} \exp\left(-\frac{z^2}{2}\right) \prod_{i=1}^{d} \left(\Phi\left(\frac{\ddot{\tau}_i}{\sqrt{1-r_i^2}}\right) + \frac{r_i z}{\sqrt{1-r_i^2}}\phi\left(\frac{\ddot{\tau}_i}{\sqrt{1-r_i^2}}\right)\right) dz \tag{38}$$

$$\approx \prod_{i=1}^{d} \Phi\left(\frac{\ddot{\tau}_i}{\sqrt{1-r_i^2}}\right) \int_{z=-\infty}^{z_{\min}} \frac{(z_{\min}-z)^p(1+za)}{\sqrt{2\pi}} \exp\left(-\frac{z^2}{2}\right) dz \ ,$$

where

$$a = \sum_{i=1}^{d} \frac{r_i}{\sqrt{1-r_i^2}} \frac{\phi\left(\frac{\ddot{\tau}_i}{\sqrt{1-r_i^2}}\right)}{\Phi\left(\frac{\ddot{\tau}_i}{\sqrt{1-r_i^2}}\right)} \ . \tag{39}$$

It finally comes

$$\text{deriv-EI}_N(\boldsymbol{x}) \approx \text{LikelyMin}(\boldsymbol{x}) \times \text{cond-EI}^{(p)}(\boldsymbol{x}), \tag{40}$$

with

$$\text{LikelyMin}(\boldsymbol{x}) := \text{Vol}(\mathcal{E}(\boldsymbol{x},\varepsilon)) f_{\partial Y_N(\boldsymbol{x})}(\boldsymbol{0}) \prod_{i=1}^{d} \Phi\left(\frac{\ddot{\tau}_i}{\sqrt{1-r_i^2}}\right)$$

$$= v\varepsilon^d \times \exp\left(-\frac{\dot{\boldsymbol{m}}^T \dot{\boldsymbol{S}}^{-1} \dot{\boldsymbol{m}}}{2}\right) \times \prod_{i=1}^{d} \Phi\left(\frac{\ddot{\tau}_i}{\sqrt{1-r_i^2}}\right), \tag{41}$$

where $v$ is a constant independent of $\boldsymbol{x}$ and $\varepsilon$, and with

$$\text{cond-EI}^{(p)}(\boldsymbol{x}) := \begin{cases} s\left((z_{\min}-a)\Phi(z_{\min}) + \phi(z_{\min})\right) & \text{if } p=1, \\ s^2\left((1+z_{\min}^2 - 2az_{\min})\Phi(z_{\min}) + (z_{\min}-2a)\phi(z_{\min})\right) & \text{if } p=2. \end{cases} \tag{42}$$

The expression when the constraint on the second order derivatives is not considered can be recovered by making $\ddot{\tau}_i$ tend to infinity. In this case, $a$ tends to 0 and $\prod_{i=1}^{d} \Phi\left(\frac{\ddot{\tau}_i}{\sqrt{1-r_i^2}}\right)$ tends to 1.

In summary, the approximation provided by Eq. (40) is based on the following four assumptions:

- the off-diagonal terms of $\partial^2 \ddot{Y}_N$ can be neglected,
- the value of $\varepsilon$ is sufficiently small for $f_{\partial Y_N(\boldsymbol{x})}(\dot{\boldsymbol{y}})$ to be approximated by $f_{\partial Y_N(\boldsymbol{x})}(\boldsymbol{0})$ for any $\dot{\boldsymbol{y}}$ in $\mathcal{E}(\boldsymbol{x}, \varepsilon)$,
- the components of $D^2 Y_N(\boldsymbol{x})$ conditioned by the event $(\partial Y_N(\boldsymbol{x}) = \boldsymbol{0}, Y_N(\boldsymbol{x}) = y)$ are statistically independent,
- the first order Taylor expansion of the function $\Phi$ provided in Eq. (37) holds.

**Numerical assessment of the proposed approximation**

In order to numerically evaluate the quality of the approximation of deriv-$\text{EI}_N$, we implement a second Monte-Carlo based approximation :

$$\text{deriv-EI}_N(\boldsymbol{x}) \approx \widehat{d}(\boldsymbol{x}) := \text{Vol}(\mathcal{E}(\boldsymbol{x}, \varepsilon)) f_{\partial Y_N(\boldsymbol{x})}(\boldsymbol{0}) \frac{1}{M} \sum_{m=1}^{M} 1_{Y_m \leqslant y_{\min}} 1_{\ddot{\boldsymbol{Y}}_m \in \mathcal{M}^+(d)} (y_{\min} - Y_m), \qquad (43)$$

where $\left\{Y_m, \ddot{\boldsymbol{Y}}_m\right\}_{m=1}^{M}$ gathers $M$ independent realizations of $(Y_N(\boldsymbol{x}), \partial^2 Y_N(\boldsymbol{x}))|\partial Y_N(\boldsymbol{x}) = \boldsymbol{0}$. The term $\text{Vol}(\mathcal{E}(\boldsymbol{x}, \varepsilon)) f_{\partial Y_N(\boldsymbol{x})}(\boldsymbol{0})$ is explicitly calculated in the same way in both approximations.

Focusing on the test functions listed in Section 4.2 in dimensions $d \in \{2, 3, 5\}$, and limiting ourselves to $N \in \{2d, 5d, 10d\}$, we uniformly sample 1000 points $\boldsymbol{x}_1, \ldots, \boldsymbol{x}_{1000}$ in $[0,1]^d$. We compute the coefficient of determination $R^2$ between the values of $\left\{\text{LikelyMin}(\boldsymbol{x}_j) \times \text{cond-EI}^{(p)}(\boldsymbol{x}_j)\right\}_{j=1}^{M}$ (the proposed and fast approximation of deriv-$\text{EI}_N$) and the values of $\left\{\widehat{d}(\boldsymbol{x}_j)\right\}_{j=1}^{M}$ (the time consuming Monte-Carlo approximation of deriv-$\text{EI}_N$). The mean value and standard deviation of these coefficients of determination obtained when repeating 10 times the whole process are finally summarized in Table 1. In each case, we observe high coefficients of determination, which justifies the use of the fast approximation.

## B Expression of the analytical test functions

The functions $y^{1\text{D}}$ and $y^{2\text{D}}$ which were considered in Section 4.1 have the following expressions (notice the offset made such that the minimum of the functions is 0):

$$y_0^{1\text{D}} : \begin{cases} [0,1] & \to & \mathbb{R} \\ x & \mapsto & \cos(6\pi x + 0.4) + (x - 0.5)^2 \end{cases}$$

$$y^{1\text{D}}(x) = y_0^{1\text{D}}(x) - \min_{z \in [0,1]} y_0^{1\text{D}}(z), \ x \in [0,1],$$

$$y_0^{2\text{D}} : \begin{cases} [0,1]^2 & \to & \mathbb{R} \\ (x_1, x_2) & \mapsto & 10 + x_1 + \left(15 x_2 - \frac{5(15 x_1 - 5)^2}{(4\pi^2} + \frac{5(15 x_1 - 5)}{\pi} - 6\right)^2 + 10 \cos(15 x_1 - 5)\left(1 - \frac{1}{8\pi}\right) \end{cases}$$

| $d$ | $\theta$ | $N$ | mean of $R^2$ | standard deviation of $R^2$ |
|---|---|---|---|---|
| 2 | 0.2 | 4 | 0.94 | 0.04 |
| 2 | 0.5 | 4 | 0.96 | 0.03 |
| 2 | 0.2 | 10 | 0.94 | 0.02 |
| 2 | 0.5 | 10 | 0.95 | 0.02 |
| 2 | 0.2 | 20 | 0.95 | 0.02 |
| 2 | 0.5 | 20 | 0.98 | 0.02 |
| 3 | 0.2 | 6 | 0.96 | 0.02 |
| 3 | 0.5 | 6 | 0.96 | 0.06 |
| 3 | 0.2 | 15 | 0.95 | 0.01 |
| 3 | 0.5 | 15 | 0.98 | 0.02 |
| 3 | 0.2 | 30 | 0.96 | 0.02 |
| 3 | 0.5 | 30 | 0.98 | 0.01 |
| 5 | 0.2 | 10 | 0.93 | 0.04 |
| 5 | 0.5 | 10 | 0.97 | 0.03 |
| 5 | 0.2 | 25 | 0.92 | 0.02 |
| 5 | 0.5 | 25 | 0.96 | 0.03 |
| 5 | 0.2 | 50 | 0.94 | 0.01 |
| 5 | 0.5 | 50 | 0.95 | 0.06 |

Table 1: Coefficients of determination $R^2$ between the proposed, fast, approximation of deriv-EI$_N$ and a Monte-Carlo approximation.

$$y^{\mathrm{2D}}(\boldsymbol{x}) = y_0^{\mathrm{2D}}(\boldsymbol{x}) - \min_{\boldsymbol{z} \in [0,1]^2} y_0^{\mathrm{2D}}(\boldsymbol{z}), \ \boldsymbol{x} \in [0,1]^2.$$

## C  Generation of test functions as realizations of a Gaussian process

We describe here how deterministic functions are defined as particular realizations of a centered Gaussian process $Z$ indexed by $\boldsymbol{x} \in \mathbb{X}$, where $\mathbb{X}$ is a compact subset of $\mathbb{R}^d$ and the covariance function of $Z$ is denoted by $C$. The process consists of the following steps.

1. We start by generating a design of experiments $\mathcal{X} := \{\boldsymbol{x}_1, \ldots, \boldsymbol{x}_N\}$ of large size $N$ covering as much as possible $\mathbb{X}$.

2. We then project $Z$ on this design of experiments, and we note $\boldsymbol{z} = (z_1, \ldots, z_N)$ the vector containing the values of $Z$ realized at $\boldsymbol{x}_1, \ldots, \boldsymbol{x}_N$.

3. By Gaussian conditioning, the function $\boldsymbol{x} \mapsto \boldsymbol{r}(\boldsymbol{x})^T \boldsymbol{R}^{-1} \boldsymbol{z}$ can thus be seen as the continuous extension of a realization of $Z$ whose projection in $\mathcal{X}$ is equal to $\boldsymbol{z}$, where for all $\boldsymbol{x} \in \mathbb{X}$,

$$\boldsymbol{r}(\boldsymbol{x}) = \begin{pmatrix} C(\boldsymbol{x}, \boldsymbol{x}_1) \\ \vdots \\ C(\boldsymbol{x}, \boldsymbol{x}_N) \end{pmatrix}, \ \ \boldsymbol{R} := \begin{bmatrix} C(\boldsymbol{x}_1, \boldsymbol{x}_1) & \cdots & C(\boldsymbol{x}_1, \boldsymbol{x}_N) \\ \vdots & \cdots & \vdots \\ C(\boldsymbol{x}_N, \boldsymbol{x}_1) & \cdots & C(\boldsymbol{x}_N, \boldsymbol{x}_N) \end{bmatrix}.$$

It is clear that this type of construction strongly depends on the dimension of $\mathcal{X}$. Indeed, the larger the size of $\mathcal{X}$, the closer the constructed function will look like a particular realization of $Z$. In addition, the larger the input space dimension, $d$, the more points will be needed in $\mathcal{X}$ for the continuous extension to be relevant. For the examples treated in Section 4.2, $\mathcal{X}$ is defined as the concatenation of a two-level factorial design (in order to cover all the vertices of $\mathbb{X} = [0,1]^d$) and a space filling LHS design of size $100 \times d$.

