# OpenReview forum: "Bayesian optimization with derivatives acceleration"
_TMLR — Accepted by TMLR_

### Review · Reviewer_BQ67 · 2024-05-25

**Summary Of Contributions:**

The paper proposes deriv-EI, a BO acquisition function that is an extension of expected improvement (EI). deriv-EI selects the next input to sample based not only on the improvement score, but on the likelihood that an input is a local minima, which is the case if the sample path gradient is 0 and the Hessian is positive semidefinite. This is possible because the derivatives of a GP are also GPs. deriv-EI is a tractable approximation to these criteria. deriv-EI is shown to outperform EI on synthetic test functions of dimension 1, 2, 3 and 5.

**Audience:**

Yes

**Broader Impact Concerns:**

None.

**Claims And Evidence:**

Yes

**Requested Changes:**

### Would strengthen work:
1. Include real world experiments.
2. Run experiments with noise.
3. A more complete discussion on related works also using GP derivatives.

**Strengths And Weaknesses:**

### Strengths:
1. The central claim of the paper is met: deriv-EI appears to empirically outperform EI on the provided test functions.
2. The idea is interesting and makes sense. There seems to be potential for this technique to be more generally applicable to other acquisitions beyond EI.
3. The exposition is clear.

### Weaknesses:
1. For an algorithm with no theoretical guarantees, the empirical evaluation is somewhat narrow. All the test functions are synthetic and of low dimension. The empirical evaluation would be stronger if it included real world experiments, for example, hyperparameter tuning.
2. While EI was originally proposed as for the noiseless evaluations setting, the modern BO research landscape almost always considers the case of noisy evaluations in which the observed values $y = f(x) + \epsilon$ where $\epsilon$ is a noise random variable. The proposed algorithm's practical usefulness is less proven when the experiments are run with noiseless observations.
3. deriv-EI assumes that the global minima is within the interior of the search space, as opposed to on the boundary, an assumption that EI does not make and has to be handled as discussed in Sec 5.1. Furthermore, deriv-EI depends on an additional hyperparameter $\epsilon$ which controls the size of the ellipsoid under which a gradient is taken to be sufficiently small. These additional complications provide a less compelling case for deriv-EI to be a drop-in replacement for EI. Can the authors explain the statement "The precise choice of ε has thus no impact"? Surely if we take $\epsilon$ to be very large, the gradient condition disappears?
4. A more complete discussion on related works also using GP derivatives would be helpful, e.g., explaining if this paper is related to "Bayesian Optimization with Gradients" (Wu et al., 2017) and other related papers.

---

### Review · Reviewer_QSQ4 · 2024-05-26

**Summary Of Contributions:**

The authors propose a variant of the well-known expected improvement (EI) acquisition function for (Gaussian-process-based) Bayesian Optimization. The proposed acquisition function restricts the original EI acquisition function only to consider function values of the GP posterior near local minima, i.e., having an almost vanishing gradient and a positive-definite Hessian matrix. The authors then introduce an approximation of said objective that avoids the necessity of Monte-Carlo approximations for evaluating the acquisition function. Subsequently, they compare the resulting BO method to the EI baseline, showing accelerated minimization of the chosen evaluation functions.

**Audience:**

Yes

**Broader Impact Concerns:**

No broader impact concerns

**Claims And Evidence:**

Yes

**Requested Changes:**

* Minor: I would welcome legends in all Figures such that a reader does not need to search for the meaning of a line in the text caption.
* Minor: There is a broken reference on Page 12
* Minor: The statement that deriv-EI leads to lower, better values of $y^{1D}$ in Figure 2 for intermediate values of k seems overstated given the plot in Figure 2. Also why is Figure 2(a) in log-space compared to the others?
* Minor: Could the authors clarify why they normalize the gradient with the square root of the covariance in Eq (12)?

* The paper would benefit from a more extended discussion on why only looking at the minima of the "imagined" functions should improve optimization behavior. Looking at Figure 1(a), the number/location of the GP minima does not need to correlate with the minima of the true function and consequently, focusing only on the local minima of the GP may improve or degrade performance.
* I would like to see more evaluations of the approximation of the deriv-EI objective. Table 1 shows that the objective works reasonably well for a low number of function samples but does not show how it evolves with an increasing number of samples. Given that, e.g., Figure 2(d) shows high levels of improvement of deriv-EI over EI for around N=13 data points on the corresponding two-dimensional function, it would be interesting to see how good the approximation is for this number of data points. I recommend that the authors add a hindsight analysis of the conducted experiments by comparing the approximated deriv-EI values with MC approximations for the generated optimization traces, thereby showing the approximation quality deriv-EI for the conducted optimizations. Additionally, it would be insightful to have an "adversarial scenario", in which the approximation clearly fails. I imagine that this happens with strong off-diagonal hessian terms, given that these are ignored in the approximation of deriv-EI.
* Finally, I believe that adding the following experiments to Section 5 would strengthen the overall paper:
    * An evaluation on functions for which the maxima are on the border of the search space. Although the authors clearly state that the method is not well-suited for these scenarios, it would be interesting to see whether and how much worse it is compared to EI.
    * An evaluation on a real-world task, e.g., the [(surrogate) SVM task](https://github.com/automl/HPOBench/blob/master/hpobench/benchmarks/surrogates/svm_benchmark.py) used in [1]


[1] Falkner, S., Klein, A., & Hutter, F. (2018, July). BOHB: Robust and efficient hyperparameter optimization at scale. In International conference on machine learning (pp. 1437-1446). PMLR.

**Strengths And Weaknesses:**

Strengths:
* The paper is well organized and written. As someone being familiar with the topic of BO but being rather an "end-user" of the methods, I had no issues following the paper.
* I enjoyed the open discussion about the assumptions and limitations of the proposed method as well as the emphasis on it realizing a different tradeoff between exploration and exploitation than EI.
* From a technical side, I enjoyed the use of the joint distribution over function values and first-and second-order derivatives.

Weaknesses:
* The proposed approach's motivation mostly stems from intuition, with no formal or statistical reason for why incorporating derivatives of GPs should improve performance (I'll detail that below).
* The paper also lacks additional evaluations on other functions (detailed below) and an extended discussion of how and when the derived approximation to the deriv-EI objective fails.

---

### Review · Reviewer_Hpa1 · 2024-05-31

**Summary Of Contributions:**

This work proposes the use of derivative & Hessian estimate of the target function in its acquisition criterion for BO.  It provides an approximation of such criterion, and show that it indeed helps finding better estimate quicker than the one without those information on toy examples.

**Audience:**

Yes

**Claims And Evidence:**

Yes

**Requested Changes:**

1. Please add more detailed step-by-step explanations on Appendix A.  Especially, although I might be missing something here, please define $m$, $\dot{x}$, etc. etc. and please elaborate the approximate equality on (34) and (35) more so that I can easily verify the correctness.

2. Please motivate the focus on the class of kernels used in this work and on EI; also please confirm that the experimental results are not very specific but that it should be the case this advantage is seen in wider domains of problems.

3. Figure 1 is a bit hard to parse; please provide a bit more detailed explanation

Minor points:
1. In abstract, it says "dimension 2, 3, and 5" and it is a bit confusing why these specific numbers are important; otherwise, should say "low dimension problems" or so.
2. a d-dimensional real space -> the?
3. 4th row introduction:  the parenthesis seems not to be closed
4. What do you mean by "BO is difficult to beat"?  it sounds a bit strange
5. Please confirm that you use $C^2$ as twice differentiable function class, not twice continuously differentiable.
6. What is "MatÃľrn kernels"?  Matern?
7. in page 6  "A" -> "Appendix A"
8. in 12 what do you mean by "These functions are call"?
9. in 12   "Section ??" please fix this
10. in page 13  "will typicallybe" -> "will typically be"
fix other minor typos as well.

**Strengths And Weaknesses:**

Overall, this is a good and solid paper I think, if some of the concerns are addressed.

Strength:
1. The idea is very simple yet very general and practically useful.
2. The paper is overall very well written and the idea is clear.
3. It talks about some limitations too (the case the minima are on the edge etc.)

Weakness:
1. The major weakness is the lack of detailed step-by-step explanations on the derivation of approximation (especially Appendix A)
2. Why focusing on certain kernels, EI etc. are not very well motivated

Minor weakness:
1. Some typos
2. Clarity of figures

---

### Decision · Action_Editor_iGGJ · 2024-07-24

**Recommendation:** Accept with minor revision

**Comment:**

All the reviewers found that the claims were justified by evidence provided.  The ratings provided by reviewers, however, were somewhat mixed with one Accept, one Leaning Accept and one Leaning Reject.  Multiple reviewers leaned towards reject before the discussion period.  One of them seems to have been satisfied by the author response.  The other remaining reviewer cited the lack of response as a reason for their Leaning Reject rating, though they noted that they wouldn't be strongly opposed to acceptance.  The authors have since responded, but after the reviewer posted their decision (and they haven't checked the response since - nor are they required to given the timeline).  It seems that the authors addressed most of the reviewers' concerns within the response period and have made substantive changes to their manuscript.  Having checked the authors' response, it seems that they have satisfied the concerns of the remaining negative reviewer.  I appreciate that the authors updated the manuscript and highlighted the changes since the submission.  Therefore, and given the two accepts, I am recommending accept with minor revisions.  The minor revisions include the proposed changes from the rebuttal. Most of these are already worked into the manuscript, but obviously they should no longer be highlighted for the camera ready version.  Given the one reviewers' concerns, I would like to verify that these changes have made it into the camera ready.

**Audience:**

The reviewers all said that the work would be interesting to a subset of the community.  Bayesian optimization is a highly active subfield within machine learning and prevalent in scientific design applications.  Thus, the use case where derivates are available and could accelerate the optimization procedure could be of interest.

**Claims And Evidence:**

The reviewers all found that the claims were supported by evidence.